# A virus responds instantly to the presence of the vector on the host and forms transmission morphs

**Alexandre Martinière[1†], Aurélie Bak[1†], Jean-Luc Macia[1], Nicole Lautredou[2], Daniel Gargani[1], Juliette Doumayrou[1,3], Elisa Garzo[4], Aranzazu Moreno[4], Alberto Fereres[4], Stéphane Blanc[1]\*, Martin Drucker[1]\***

[1]Virus Insect Plant Laboratory, INRA, Mixed Research Unit 385, Campus International de Baillarguet, Montpellier, France; [2]MRI Imaging Platform, Institute of Human Genetics, Campus CNRS Arnaud de Villeneuve, Montpellier, France; [3]Infectious Diseases and Vectors: Ecology, Genetics, Evolution and Control Laboratory, CNRS UMR 5290, Montpellier, France; [4]Department of Crop Protection, CSIC, Institute of Agricultural Sciences, Madrid, Spain

**Abstract** Many plant and animal viruses are spread by insect vectors. Cauliflower mosaic virus (CaMV) is aphid-transmitted, with the virus being taken up from specialized transmission bodies (TB) formed within infected plant cells. However, the precise events during TB-mediated virus acquisition by aphids are unknown. Here, we show that TBs react instantly to the presence of the vector by ultra-rapid and reversible redistribution of their key components onto microtubules throughout the cell. Enhancing or inhibiting this TB reaction pharmacologically or by using a mutant virus enhanced or inhibited transmission, respectively, confirming its requirement for efficient virus-acquisition. Our results suggest that CaMV can perceive aphid vectors, either directly or indirectly by sharing the host perception. This novel concept in virology, where viruses respond directly or via the host to the outside world, opens new research horizons, that is, investigating the impact of 'perceptive behaviors' on other steps of the infection cycle.

**\*For correspondence:** blanc@ supagro.inra.fr (SB); drucker@ supagro.inra.fr (MD)

[†]These authors contributed equally to this work

**Competing interests:** The authors declare that no competing interest exist

## Introduction

Transmission is a pivotal step in the infection cycle of viruses: it controls the passage from one host to another and is thus essential for dissemination. This step can represent a significant bottleneck for the infection cycle, since it is common for only a small proportion of the countless viral genomes produced to be passed on to a new host in a transmission event; for example, only one to three of the many transmissible genomes initiate a new infection after *Potato virus Y* transmission (*Moury et al., 2007*). It is thus expected that viruses have adapted their life cycle and developed sophisticated strategies to optimize their transmission. Whereas non-viral pathogens are known to allocate resources for the production of transmission-specific morphs (discussed in *Matthews, 2011*), surprisingly little is known for this mechanism regarding viruses (for review see *Blanc et al., 2011*). Some viruses are transmitted vertically to host offspring and others are transmitted by contact between hosts (e.g., by wind, water or physical contact), but most viruses rely on vectors for rapid proliferation within host populations (for review see *Kuno and Chang, 2005*; *Blanc et al., 2011*; *Bak et al., 2012*).

The most important vectors are found among the arthropods. Those with a piercing-sucking feeding behavior such as mosquitoes (or other blood-feeding dipterans) and ticks are especially significant for vertebrate viruses, and likewise aphids, white flies and other sap-feeding bugs are consequential

**eLife digest** Viruses are infectious agents that can replicate only inside a living host cell. When a virus infects an animal or plant, it introduces its own genetic material and tricks the host cells into producing viral proteins that can be used to assemble new viruses. An essential step in the life cycle of any virus is transmission to a new host: understanding this process can be crucial in the fight against viral epidemics.

Many viruses use living organisms, or vectors, to move between hosts. In the case of plant viruses such as cauliflower mosaic virus, the vectors are often aphids. When an aphid sucks sap out of a leaf, virus particles already present in the leaf become attached to its mouth, and these viruses can be transferred to the next plant that the insect feeds on. However, in order for cauliflower mosaic virus particles to become attached to the aphid, structures called transmission bodies must form beforehand in the infected plant cells. These structures are known to contain helper proteins that bind the viruses to the mouth of the aphid, but the precise role of the transmission body has remained obscure.

Now Martinière et al. show that the transmission body is in fact a dynamic structure that reacts to the presence of aphids and, in so doing, boosts the efficiency of viral transmission. In particular, they show that the action of an aphid feeding on an infected leaf triggers a rapid and massive influx of a protein called tubulin into the transmission body. The transmission body then bursts open, dispersing helper protein–virus particle complexes throughout the cell, where they become more accessible to aphids. This series of events increases viral transmission rates twofold to threefold.

The results show that a virus can detect insect vectors, likely by using the sensory system of its host, and trigger a response that boosts viral uptake and thus transmission. This is a novel concept in virology. It will be important to discover whether similar mechanisms are used by other viruses, including those that infect animals and humans.

for plant viruses. These vectors are ideal, as their variety of mouth parts can puncture cells, blood vessels, and plant sap vessels with great precision, thus enabling efficient uptake and injection of pathogens without killing the host. Vector transmission can be classified into two main transmission modes. In circulative transmission, the virus is taken up by the vector together with the nutrients (e.g., blood, plant sap, cell contents), where it actively crosses from the intestine into the vector interior. Then it cycles through the hemocoel (the internal body cavity awash in hemolymph) to the salivary glands, where the virus can be secreted together with the saliva into a new host. The second transmission mode is alternatively referred to as mechanical transmission (for human and animal viruses) or non-circulative transmission (for plant viruses). In this transmission mode, the arthropod vector only briefly comes into contact with the virus, in which it transiently attaches to the vector mouth parts and is subsequently released; an internalization step does not occur. The viral proteins involved in this seemingly simple process have been well-described in the literature, often down to the molecular level (for review see *Ng and Falk, 2006*). On the other hand, their precise roles during virus-acquisition by the vector remains largely unexplored (for review see *Blanc et al., 2011*).

*Cauliflower mosaic virus* (CaMV), the virus studied here, is a non-circulative virus transmitted by aphids. CaMV binds to a receptor protein located at the tip of the aphid's needle-like mouth parts, the stylets (*Uzest et al., 2007*, *2010*). The CaMV transmissible complex is composed of the icosahedral viral particle (containing the viral genome enclosed by a shell of capsid protein P4), the virus-associated protein P3, and finally the aphid-transmission factor or helper component, the viral protein P2 (*Blanc et al., 1993a*; *Leh et al., 1999*; *Plisson et al., 2005*). P2 is central to the virus's transmission, as it links the virus particle to the aphid stylets through the interaction of its C-terminus with virus-associated P3, as well as the linking of its N-terminus with the stylet receptor (*Figure 1A*). Interestingly, although P2 deletion mutants of CaMV are not transmissible by aphids, they are perfectly infectious when inoculated artificially to host plants. This shows that the only role for P2 in the CaMV life cycle is virus–vector interaction. P2 localizes exclusively to a specific cytoplasmic inclusion in infected plant cells, the transmission body (TB, *Figure 1B*). There, P2 co-aggregates with the viral protein P3 to form a matrix in which some virus particles are embedded; the existence of any cellular components within this matrix remains elusive (*Espinoza et al., 1991*; *Drucker et al., 2002*). The TB-contained P3 is most likely

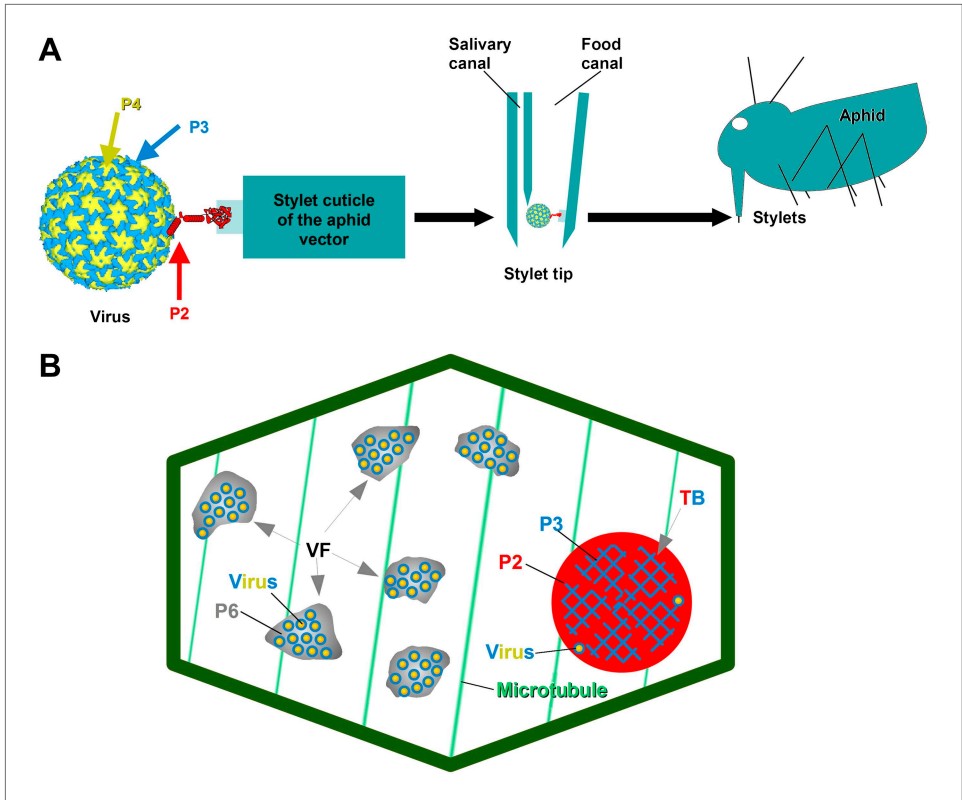

**Figure 1**. The CaMV transmissible complex and the transmission body. (**A**) Left: the CaMV transmissible complex comprises the virus particle, composed of capsid protein P4 (yellow), virus-associated protein P3 (blue) and the helper component P2 (red). P2 binds via its C-terminus to P3 and via its N-terminus to a protein receptor localized in the stylet tips of the aphid vector (middle and right). (**B**) Infected cells contain many cytoplasmic virus factories (VF), where most virus particles (blue-yellow circles) accumulate in a matrix composed of virus protein P6 (grey), and a single transmission body (TB). The TB (also cytoplasmic) is composed of P2 (red) and P3 (blue) as well as scattered virus particles. P3 in TBs is most likely in a conformation that differs from virus-associated P3. The spatial separation of the components of the transmissible complex (P2 in the TB and most virus particles in VFs) lead us to propose that they unite only at the moment of vector acquisition (***Drucker et al., 2002***). Cortical microtubules are designated in green and the cell wall in dark green. Cell organelles are not shown, for clarity. The CaMV model is from ***Plisson et al. (2005)***.

dissimilar in conformation to the P3 associated with the virus particle, and has been suggested to play a role in TB structure and maintenance, but any details are yet unknown (***Drucker et al., 2002***; ***Hoh et al., 2010***). TBs are indispensable to this transmission, as it has previously been shown that aphids are unable to acquire the virus in their absence (i.e., P2 deletion mutants, ***Woolston et al., 1983***), as well as when TBs are malformed. The P2 mutant, P2$_{G94R}$ (described in ***Khelifa et al., 2007***), assists efficiently in the transmission of purified CaMV particles associated with P3, when aphids are allowed to acquire all three components in vitro from suspensions across Parafilm membranes. However, when the P2$_{G94R}$ mutant is expressed *in planta* in the context of CaMV infection, it induces the formation of a misshaped TB (for details see ***Khelifa et al., 2007***), preventing plant-to-plant transmission by the aphid vector.

To understand CaMV acquisition, some knowledge of the unique feeding behavior of aphid vectors is required. Aphids feeding on plant leaves insert their stylets into the middle lamella that separates adjacent cells, and subsequently perform a series of brief test punctures into the epidermis and parenchyma cells; this continues until eventually reaching the phloem where they can feed for long periods, provided the plant is a suitable host (for review see ***Fereres and Moreno, 2009***). Most plant cells survive the initial test punctures, during which only minute amounts of cytoplasm are ingested. CaMV and hundreds of other viral species can be acquired during this feeding behavior, but detailed events

occurring within the punctured cell at the precise moment of stylet entry and how they result in virus acquisition are largely unknown. In the example of CaMV, it is known that microtubules are involved in the generation of TBs at the onset of infection (*Martinière et al., 2009*) and that the microtubule depolymerizing drug oryzalin inhibits virus acquisition by aphids (*Martinière et al., 2011a*), but any details of the mode of action of oryzalin on TBs and how TBs function in virus transmission are still unclear.

Here, we have analyzed the three-way interaction between CaMV, host plant cell, and aphid at the precise moment of the intracellular penetration of the stylets and imminent virus acquisition. Our study reveals an unforeseen capability of CaMV in that it senses—probably by using the host cell machinery—the aphid feeding, and then instantly produces a transmissible form for uptake by the insect.

## Results

### Different TB forms are detected in infected plant cells

At the beginning of this study was the observation that several different TB phenotypes could be discerned in infected tissues, as viewed by double-labeling experiments using antibodies against the TB marker P2 and the microtubule protein α-tubulin. Thus, typical TBs were detected, and these were rather large (2–5 μm in diameter) and mostly ovoid single cytoplasmic inclusions, having a cortex heavily labeled by P2 antibody and a less intensely labeled interior. Most importantly, little if any tubulin was detected in this regular form of the TB (*Figure 2A*). Interestingly, we also observed a second class of TBs that was phenotypically nearly identical, with the exception that tubulin had greatly accumulated in their centers (*Figure 2B*). Even more surprisingly, we were unable to detect TBs in some cells; at best, small P2 foci without the typical TB structure were visible (hereafter referred to as fragmented TBs). Instead, most P2 decorated the microtubule network in these cells (a P2 distribution pattern hereafter designated as 'mixed-networks' and referring to mixed P2-tubulin networks; *Figure 2C*). A common point among these three TB phenotypes was that their occurrence varied greatly from one experiment to the next. Depending on the tissue preparation, anywhere from almost none to practically all of the TBs contained tubulin; the proportion of cells containing mixed-networks also varied from one experiment to another. To account for these observations, we hypothesized that, among the different TB morphologies observed (hereafter referred to as 'morphs'), the tubulin-loaded (Tub$^+$|TB) and mixed-network phenotypes were induced by unidentified stresses during leaf handling, whereas the tubulin-less phenotype (Tub$^-$|TBs) corresponded to unstressed 'standby' TBs found under normal conditions. As the only known role for TBs is in transmission, this further raised the question of whether and how the presence of tubulin within TBs impacts transmission by aphids. To investigate this phenomenon, we first aimed to identify artificial stresses that could trigger specific transformation of standby TBs into Tub$^+$|TBs and mixed-networks. Heat-shock, wounding and $CO_2$ exposure all induced TB transformation (*Figure 2D–F* and *Figure 2—figure supplement 1*). Quantification of the various TB morphs (*Figure 3*) established that all three treatments induced Tub$^+$|TBs. However, mixed-networks (co-existing with Tub$^+$|TBs and fragmented TBs) were significantly increased only in wounded or $CO_2$-exposed leaves, and were rarely observed in heat-shocked or untreated leaves. This indicates that the different stresses had different and specific effects on TBs.

Having access to an experimental system to specifically induce TB transformation, we next examined what role(s) the different TB morphs might play in CaMV aphid-transmission. Here, we pursued three complementary objectives: i) to characterize the dynamics of TB transformation; ii) to investigate whether aphid feeding activity can trigger this transformation; and iii) to test whether TB transformation is required for successful CaMV transmission.

### TB dynamics occur on a fast time scale

For TB changes to have any biological relevance in CaMV acquisition by aphid vectors, they must happen fast enough to be compatible with the duration of the intracellular penetration of the aphid stylets during test probes, that is, within ~10 s. To estimate the speed of TB transformation, we first tested wounding stress by inflicting cuts with a razor blade on infected turnip leaves. The tissue was then fixed immediately (~10 s) and the TB phenotype analyzed by immunofluorescence. Tub$^+$|TBs and mixed-networks were detected readily at the wounding sites; by contrast, the cells of control tissue, or tissue wounded after fixation, predominantly harbored standby TBs (*Figure 3*). We next evaluated the kinetics of tubulin entry into TBs. The surface of a CaMV-infected Arabidopsis leaf expressing genetically tagged GFP-tubulin (*GFP-TUA6*; *Ueda et al., 1999*) was touched with a microelectrode, and the

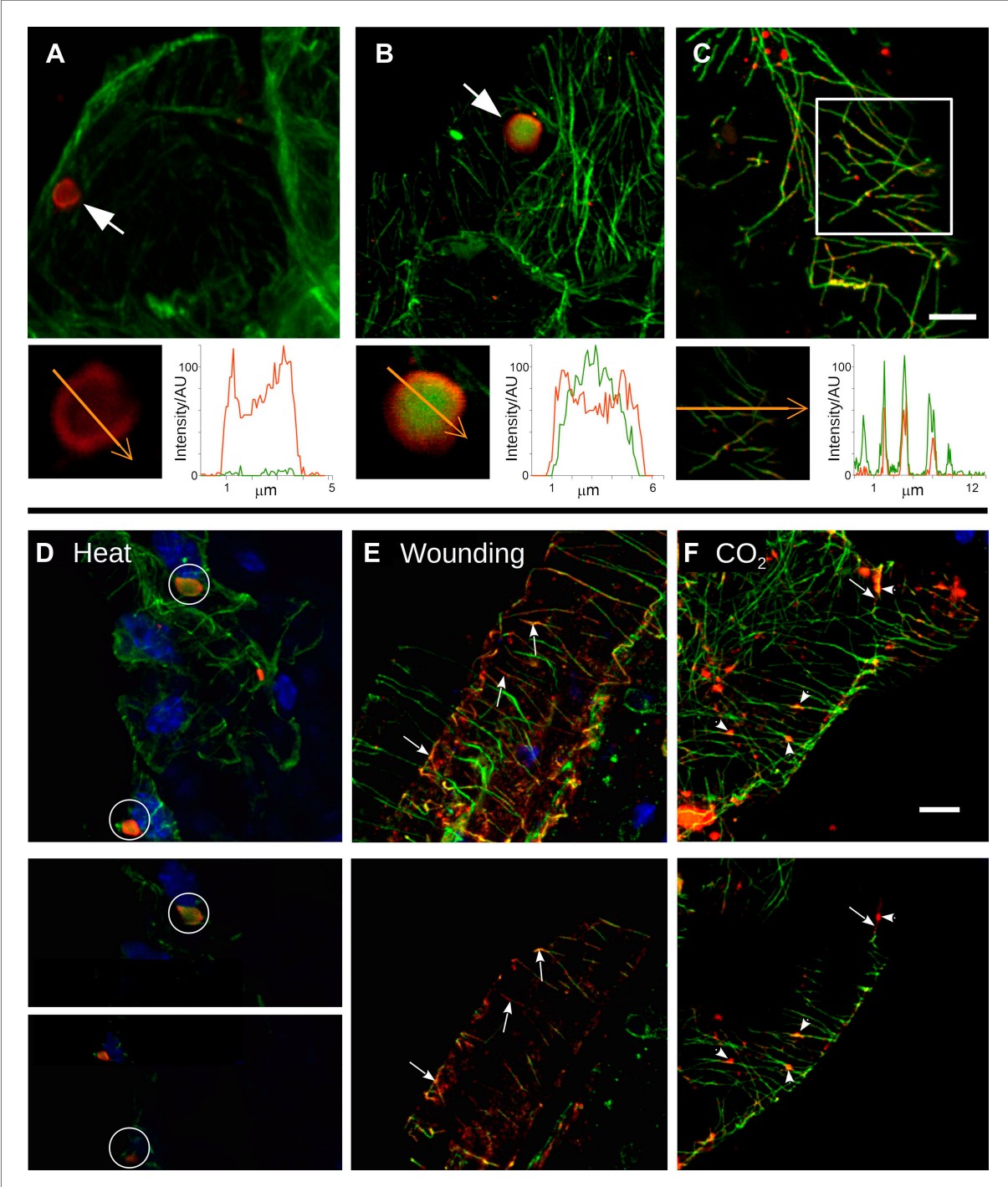

**Figure 2**. Stress induces different TB morphs. (**A**–**C**) The three TB morphs. Immunofluorescence of infected leaves against P2 (red) and α-tubulin (green), with co-labeling appearing as yellow/orange, reveals the different TB forms: (**A**) a tubulin-less TB (arrow), (**B**) a Tub⁺|TB (arrow) and (**C**) mixed-networks. Images show confocal projections; insets show optical single sections from the TBs indicated by the arrows in (**A**) and (**B**), and of the enclosed zone in (**C**). The orange arrows in the insets mark the line scans and the direction used to create the profiles of P2 (red) and tubulin (green) label intensity, shown to the right of the insets. The line scans show that the TB in (**A**) contains hardly any tubulin, whereas the TB in (**B**) is heavily tubulin-labeled, revealing stronger tubulin labeling in the center of the TB than at the cortex. Finally, the distributions of P2 and tubulin labels colocalize in the mixed-networks shown in (**C**). The intensities are indicated in arbitrary units (AU) since the acquisition conditions were not identical for the different samples. (**D**–**F**) Stress induces TB transformation. Immunofluorescence labeling (P2 in red, tubulin in green, DAPI nucleic acid stain in blue) of infected leaves after the indicated stress treatment shows that heat shock (**D**) induces only Tub⁺|TBs, whereas wounding stress (**E**) and exposure to $CO_2$ (**F**) additionally induce

*Figure 2. Continued on next page*

*Figure 2. Continued*

TB fragmentation (as revealed by the small red or orange foci in **E** and **F**) and mixed-networks. The upper panels of (**D**–**F**) show confocal projections, and the lower panels show selected optical single sections. For heat shock (**D**), two individual sections representing a median section through each of the two encircled TBs are shown. In (**E**–**F**), the arrows indicate filamentous P2 labeling that is continuous with microtubule labeling, and the arrowheads point to small P2 aggregates in the vicinity of microtubules. Scale bars: 5 μm. The confocal single sections used to create the projections shown here can be found in *Figure 2—source data 1–6*. See also *Figure 2—figure supplement 1* that shows in vivo stress response of GFP-labeled tubulin in infected plants.

The following source data and figure supplements are available for figure 2:

**Source data 1.** Confocal single sections and acquisition parameters used for *Figure 2A*.

**Source data 2.** Confocal single sections and acquisition parameters used for *Figure 2B*.

**Source data 3.** Confocal single sections and acquisition parameters for *Figure 2C*.

**Source data 4.** Confocal single sections and acquisition parameters for *Figure 2D*.

**Source data 5.** Confocal single sections and acquisition parameters for *Figure 2E*.

**Source data 6.** Confocal single sections and acquisition parameters for *Figure 2F*.

**Figure supplement 1**. Tubulin accumulation in large inclusions after different stresses is specific to TBs.

response of TBs in epidermis cells was recorded using confocal time-lapse macroscopy. In the event of a fast entry of tubulin into TB, we should expect to observe a rapid appearance of fluorescent foci, corresponding to Tub$^+$|TBs within these cells. *Figure 4A* and *Movie 1* reveal the detection of GFP-tubulin in TBs as early as ~5 s after microelectrode impact; fluorescence in these inclusions reached a maximum and stabilized within ~10 s. Similar results were obtained in ~50% of all infected cells tested (*Table 1*). Contrarily, GFP-tubulin formed a diffuse fluorescent cloud at the impact site in healthy control cells, in line with previous reports (*Hardham et al., 2008*); fast appearance of tubulin inclusions as in infected cells was never observed. Subsequently, we examined whether the tubulin within TBs is exchanged with that of the cytoplasm, by measuring GFP-tubulin turnover in Tub$^+$|TBs in fluorescence recovery after photobleaching experiments (FRAP). Photo-bleached GFP-tubulin in TBs was rapidly replaced by fresh cytoplasmic GFP-tubulin (*Figure 4B–D*), suggesting that this protein circulates continuously between Tub$^+$|TB and the cytoplasm. Taken together, these results indicate that the appearance of both Tub$^+$|TBs and mixed-networks is fast enough to occur during an aphid puncture, and that there is a dynamic equilibrium between cytosolic and TB-contained tubulin.

## TB transformation follows a precise temporal order

The above experiments demonstrating the existence of distinct TB morphs, we next turned our attention to the possible transformation of one form into another, and aimed to establish a chronology of TB morphological changes. To facilitate tracking, we used infected protoplasts to screen for various conditions that could induce TB transformation (*Table 2*), including various physical, chemical and biological stresses. These wide-ranging cell treatments showed that of all the tested stresses, only heat, the chemical sodium azide, compacting of cells by sedimentation and carbon dioxide ($CO_2$) induced TB transformation. As in leaves, heat treatment of protoplasts only induced Tub$^+$|TBs (data not shown), whereas $CO_2$ and azide stimulated Tub$^+$|TBs as well as mixed-networks (*Figure 5A,B*). The kinetics of TB transformations followed a precise order: first, standby TBs are loaded with tubulin, and then mixed-networks and tubulin-containing TB fragments appear, at the expense of Tub$^-$|TBs. After 5 min ($CO_2$ treatment) or 40 min (azide treatment) most cells displayed mixed-networks (*Figure 5C,D*). A most remarkable property of the TBs was their rapid reversion from the mixed-network phenotype back to tubulin-less TBs; this was provoked either after substituting normal air for $CO_2$ (*Figure 5E*), or after removing azide from the culture medium (*Figure 5F*). Moreover, it was possible to induce several consecutive rounds of TB transformation in the mixed-networks, as well as reversion to the same cell suspension, by continually relieving and resubmitting cells to stress (*Figure 5E*).

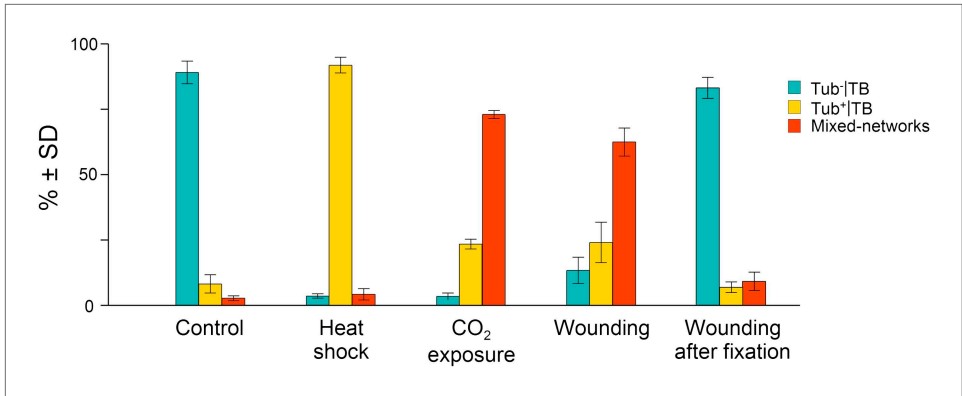

**Figure 3**. Quantitative analysis of the different TB morphs induced under stress conditions. Leaf samples were either left untreated (Control), exposed for 2 h at 37°C (Heat shock), exposed for 15 min to $CO_2$ atmosphere ($CO_2$ exposure), cut with a razor blade and then fixed within 10 s (Wounding), or fixed first and then cut with a razor blade (Wounding after fixation). All leaf samples were then processed in parallel for immunostaining against P2 and α-tubulin and scored for the occurrence of the different TB morphs: 'standby' Tub⁻|TBs (turquoise), 'activated' Tub⁺|TBs (yellow) or mixed-networks (red). Results are from three independent experiments and the total number of TBs and networks counted for each condition were control (284), heat shock (293), $CO_2$ exposure (282), wounding stress (313), and 288 in tissues wounded after fixation. See **Figure 3—source data 1** for details. SD: standard deviation.

The following source data are available for figure 3:

**Source data 1.** Source data for **Figure 3**.

## Virus particles are also redistributed onto mixed-networks

These time series experiments reveal P2 relocalization, from TBs to microtubules, following the application of different stresses. Are virus particles, the other major component of the CaMV transmissible complex, also distributed onto the mixed-networks? To answer this question, we carried out immuno-fluorescence of the mixed-networks using a CaMV capsid protein antibody. *Figure 6A* shows that capsid protein localizes to cytoplasmic inclusions (which are most likely virus factories) in unstressed cells. Under conditions that trigger the formation of the mixed-networks, that is, $CO_2$ (*Figure 6B*) or azide treatment (*Figure 6C*), P4 label colocalized with microtubules. Quantification of this observation (*Figure 6D*) indicates that almost all cells displayed P4 networks after stress treatment. Nearly the same proportion of cells contained P2 or P4 networks after treatment with $CO_2$ or azide (compare *Figure 6D* with *Figure 5C,D*), which suggested that mixed-networks are also associated with virus particles. We confirmed this by electron- and immunogold microscopy, which showed (in treated cells) that cortical microtubules displaying P2 were indeed also decorated heavily with CaMV particles (*Figure 7A–C*). Finally, we also examined the ultrastructure of standby and Tub⁺|TBs. The latter were induced by heat shock and then the tissues prepared for electron microscopy. *Figure 7D–G* shows that TBs consist of an electron-lucent matrix in which some virus particles are embedded as previously reported (*Espinoza et al., 1991*; *Drucker et al., 2002*). In control TBs, the virus particles seemed either to be distributed evenly throughout the TB matrix (*Figure 7D*) or to be more concentrated at their cortex (*Figure 7F*). Heat-shocked TBs displayed similar TB phenotypes (*Figure 7E,G*) and we could not observe any flagrant differences in TB phenotype between heat-shock-induced Tub⁺|TBs and control TBs. This corresponded to the results obtained by fluorescence microscopy (*Figure 2D,E*) where likewise no obvious differences between the two TB morphs were observed.

Taken together, these results demonstrate that mixed-network formation is always preceded by a massive entry of tubulin into the standby TB, which in itself does not induce a major ultrastructure change. However, under specific stress conditions such as azide or $CO_2$ (but not heat-shock), this tubulin entry can be followed by a total disruption of the TB. This includes an even dispersion of the components of the CaMV transmissible complex (i.e., P2 and virus particles) onto microtubules throughout the cortical cytoplasm. This redistribution might render the CaMV transmissible complexes more readily accessible to aphid vectors.

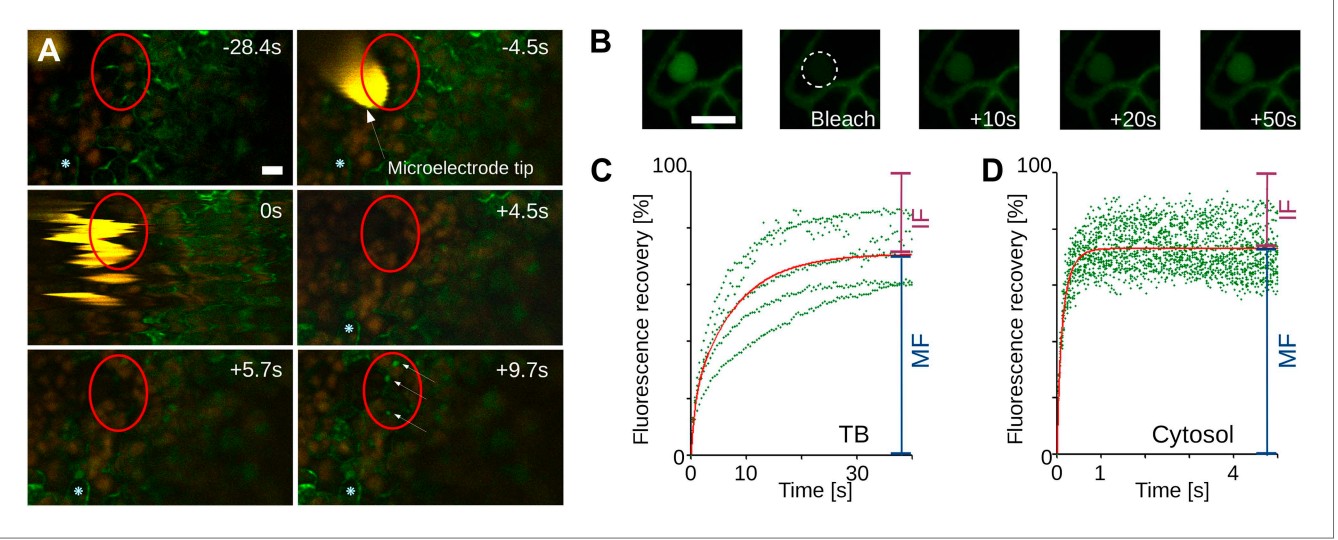

**Figure 4**. Tubulin influx into TBs occurs on a rapid time scale. (**A**) Kinetics of tubulin entry into TBs. The epidermis of CaMV-infected Arabidopsis leaves expressing GFP-tagged α-tubulin (Arabidopsis *GFP-TUA6*) was touched with a microelectrode tip (yellow), and the effect of the impact recorded by time-lapse confocal macroscopy. GFP-tubulin fluorescence is shown in green, chloroplast fluorescence in orange/red. Negative and positive time points are before and after the microelectrode-epidermis contact, respectively. The red circle denotes the impact zone, and the three arrows point to newly formed GFP-tubulin inclusions. The blue asterisk indicates a reference epidermis cell that did not change its z-position during the time lapse recording and can be used as a landmark for orientation. (**B–D**) Tubulin cycles between TBs and the cytoplasm. Arabidopsis *GFP-TUA6* plants were infected with CaMV and the epidermis was screened for rare spontaneously occurring Tub+|TBs (no deliberate stress treatment was inflicted on the leaf); these were identified by the characteristic shape of the fluorescent tubulin-containing inclusions (see *Figure 2—figure supplement 1*). The GFP-tubulin in these Tub+|TBs was photobleached, and the recovery of the GFP fluorescence (due to replacement by fresh cytoplasmic GFP-tubulin) was recorded by time lapse microscopy. (**B**) Microscopic images of a typical FRAP experiment. The first picture shows a GFP-tubulin-containing Tub+|TB before photobleach-ing. The dashed circle in the second picture indicates the photobleached zone at t = 0 s, and the following pictures show recovery of the GFP-fluorescence at indicated time points after photobleaching. (**C–D**) The graphs show quantifications of fluorescence recovery: after photobleaching of TBs (**C**), and after photobleaching of a cytoplasmic zone as a control of free tubulin diffusion (**D**). The fluorescence levels were normalized (100% = fluorescence before bleaching, 0% = fluorescence just after bleaching). For the two quantification graphs, FRAP trend lines (red) were calculated from seven FRAP experiments on GFP-tubulin-containing TBs, or from 18 FRAP experiments on cytoplasmic zones. The difference in $t_{(1/2)}$ for fluorescence recovery between TBs and the cytoplasm was highly significant (p<0.0001, t-test with n = 18 for TBs and n = 21 for cytoplasm). In contrast, the difference in the mobile fractions, that is, the percentage of exchangeable GFP-tubulin, the so-called mobile fraction, was not significant (p=0.504, t-test with n = 18 for TBs and n = 21 for cytoplasm). These results indicate that tubulin cycles between the cytoplasm and TBs, albeit at much slower rates than free diffusion in the cytoplasm. See *Figure 4—source data 1 and 2* for details. Scale bars: 10 μm; MF: mobile fraction; IF: immobile fraction.

The following source data are available for figure 4:

**Source data 1.** Source data for *Figure 4C*.

**Source data 2.** Source data for *Figure 4D*.

## Aphid behavior triggers TB transformation

The results in *Figures 3–5* show that standby TBs can transform reversibly into Tub+|TBs, and then again into mixed-networks, on a time scale that is fully compatible with aphid intracellular probing (*Fereres and Moreno, 2009*). However, in all of the above cases, TB transformations were induced by artificial stresses and might not reflect a 'natural' induction of TB changes in aphid-infested plants. We therefore aimed to determine whether aphid feeding activity by itself triggers TB transformation. Accordingly, we developed a protocol that simultaneously allows observation of the TB phenotype in infected cells and the discrimination of cells, based on whether or not they had been in direct contact with aphid stylets. For this procedure, aphids were allowed to feed on infected leaves for 15 min, before the leaf was fixed and subsequently screened by confocal microscopy. We then identified, by their auto-fluorescence, the salivary sheaths that remain in the tissue after aphid removal, and which precisely document the path followed by the stylets (*Miles, 1968*). In addition, we identified by immuno-fluorescence the different TB morphs in plant cells that were either in close contact or farther away

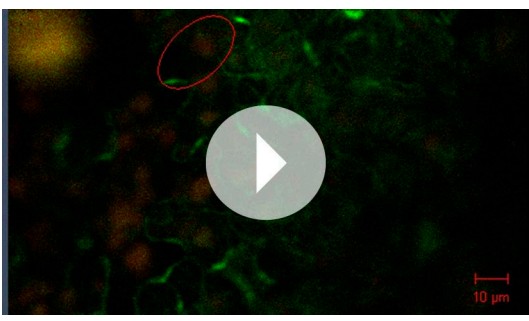

**Movie 1.** Time lapse confocal macroscopy of a leaf epidermis touched with a microelectrode tip.

**Table 1.** Speed of influx of GFP-TUA6 into TBs after touching of epidermis cells with a microelectrode

| Tubulin reaction | In infected cells | In healthy cells |
|---|---|---|
| No reaction | 14 (27%) | 3 (20%) |
| Appearance of large fluorescent inclusions within 30 s | 24 (47%) | 0 (0%) |
| Appearance of large fluorescent inclusions after >1 min | 1 (2%) | 0 (0%) |
| Formation of diffuse tubulin cloud | 12 (24%) | 12 (80%) |
| Number of experiments | 51 | 15 |

Epidermal cells of infected or healthy tissues were touched with a microelectrode and the appearance of tubulin fluorescence in inclusions was observed by time-lapse microscopy. A large fluorescent inclusion detected within 30 s or less was considered to be a wound-related tubulin entry into TB. In roughly half the experiments using infected tissue, rapid formation of large fluorescent inclusions was observed; in the other experiments, either no reaction occurred or formation of diffuse fluorescent clouds prevailed. In healthy controls, most cells responded with the appearance of diffuse tubulin clouds, as previously reported (**Hardham et al., 2008**); rapid appearance of tubulin inclusions was never observed.

from these sheaths. Standby TBs were predominantly detected in uninfested tissues, or in infested tissues greater than 15 µm from the stylet track (**Figure 8A**). In contrast, cells of infested leaves within a 15 µm perimeter of salivary sheaths often displayed typical mixed-networks, fragmented TBs and Tub$^+$|TBs (**Figure 8B,C**). Mixed-networks in cells close to the stylet track were also loaded with virus particles, as indicated by positive P4 capsid protein label of microtubules (**Figure 8D**). Quantification of the aphid-induced effect showed that 30–40% of TBs in cells in contact with a stylet track displayed a modified TB phenotype, whereas 99% of TBs in cells found more than one cell layer away from this track remained in the standby state (**Figure 8E**). In parallel experiments, aphids were removed after the 15-min feeding period and the leaves were allowed to recover for 2 h before analysis. This intriguingly provoked the aphid-induced mixed-networks to revert back to standby TBs, as demonstrated by the strong decrease in the number of modified TBs close to the stylet tracks (**Figure 8E**). This resembled the reversion observed in protoplasts, upon relief from either azide or $CO_2$ treatment (**Figure 5E,F**). Taken together, these results show that the probing activity of aphid stylets is a robust trigger of TB transformation, and that these aphid-induced TB changes are completely reversible.

## TB transformation enables efficient transmission

In order to be biologically relevant, aphid-induced TB morphs (Tub$^+$|TBs and/or mixed-networks) should display a significant impact on CaMV transmission. We therefore investigated whether TB transformation is required for aphid-transmission of CaMV. Assessing the effect of TB transformation on transmission was however not straightforward, since aphids themselves provoke TB transformation. As a possible means to circumvent this problem, we observed while quantifying TB morphs (**Figure 8E**) that only 30–40% of the TBs had transformed after contact with the stylets. We thus reasoned that a pre-treatment of infected cells that would substantially increase the proportion of modified TBs prior to aphid feeding should 'prime' these cells for virus acquisition and enhance the CaMV transmission rate. To investigate this, we first induced mixed-networks in protoplasts with azide, and then used the cells in aphid transmission experiments. We observed significantly elevated transmission rates compared to transmission from control cells that displayed mainly standby TBs (**Figure 9A**). The effect was not caused by altered aphid behavior resulting from the presence of azide in the protoplast medium, because the chemical had no effect in transmission experiments in which aphids were allowed to acquire CaMV instead from cells from suspensions containing purified virus, recombinant P2 and P3 (**Figure 9B**). Azide also had no effect on protoplast viability under the conditions used (**Figure 9C**). Taken together, these results rule out a possible confounding effect of azide, and clearly indicate that it was the presence of

**Table 2.** Effect of various treatments on TB phenotype in infected protoplasts

| Type of treatment | Treatment | Effect on TB |
| --- | --- | --- |
| Hormones | Abscisic acid [5 µM] | – |
| | Jasmonate [40 µM] | – |
| | Auxin [5 µM] | – |
| | Salicylic acid [1 mM] | – |
| Mechanical/physical stress | Compacting by sedimentation | Tub+|TB, mixed-networks |
| | Electroporation | – |
| | Heat shock | Tub+|TB |
| | Light/dark cycle | – |
| | Membrane depolarization | – |
| | Membrane hyperpolarization | – |
| | Microwaves | – |
| | Music | – |
| | Ultrasonication | – |
| | Vortexing | – |
| Elicitors | Arabinogalactan [1 mg/ml] | – |
| | Chitosan [40 µg/ml] | – |
| | Cryptogein [1 µM] | – |
| Others | $CO_2$ | Tub+|TB, mixed networks |
| | Sodium azide [0.02%] | Tub+|TB, mixed networks |
| | pH | – |

Infected protoplasts were treated/incubated under the conditions indicated, and the TB phenotype was then analyzed by immunofluorescence against P2 and α-tubulin. Protoplasts were incubated with hormones and elicitors, at the indicated final concentrations, for 60 min. Compaction of protoplasts by sedimentation was achieved by exposing them for 2 h at 9.81 m/s² on a bench-top. Electroporation conditions were 400 Ω, 0.25 µFD and 0.5 or 1 kV. Heat shock was for 1 h at 37°C. Daylight/dark cycle was for 2 h each condition. Membrane depolarization and hyperpolarization were induced with 100 and 0.1 mM KCl in protoplast buffer, respectively. Microwave exposure was 3 s at 750 W. For the music treatment (inspired by **Braam and Davis, 1990**), Vanessa Paradis's 'Joe le taxi' song was played at moderate volume (~60 db) for 3.5 min with protoplasts 'listening' from opened Eppendorf tubes. Ultrasonication consisted of a 2 s pulse at 80% power using a Bioblock Vibracell 72434 apparatus; vortexing was for 5 s at maximal power using a Vortex Genie 2 machine. Conditions for $CO_2$ and sodium azide treatments are described in 'Materials and methods'. For pH treatment, cells were incubated for 5 min with 10 mM $K_2HPO_4/KH_2PO_4$ titrated to pH 3.0, 5.6, 6.9 or 8.2. Lower and higher pH values proved lethal to the cells and were not considered for analysis. In all cases, the survival of cells was verified as described by **Widholm (1972)** and only treatments sustaining viability of the cells were used for analysis.

mixed-networks that lead to increased transmission rates. In order to compare this situation with that of intact plant tissues, we examined infected $CO_2$-treated or heat-shocked leaves in transmission assays. Under heat shock treatment, TB transformation appeared incomplete and arrested at the Tub+|TB stage, as reported above (**Figure 2D** and **Figure 2—figure supplement 1**). The heat-shocked leaves did not perform any better than controls in aphid-transmission tests (**Figure 9D**), indicating that tubulin entry into TBs alone is not sufficient to enhance transmission. In contrast, $CO_2$ induced complete TB transformation into mixed-networks (**Figure 2E**). Moreover, significantly enhanced transmission rates were recorded when $CO_2$-treated leaves were used in aphid transmission experiments (**Figure 9E**).

We conclude from these results that the sole entry of tubulin into TBs is insufficient to explain increased transmission. Furthermore, we hypothesize that the higher accessibility of CaMV to its aphid vector can be explained by microtubules serving as a scaffold for the rapid redistribution of P2 and virus particles over the entire cell. If this were true, then depolymerization of microtubules by oryzalin should both prevent the formation of mixed-networks, and significantly decrease the transmission

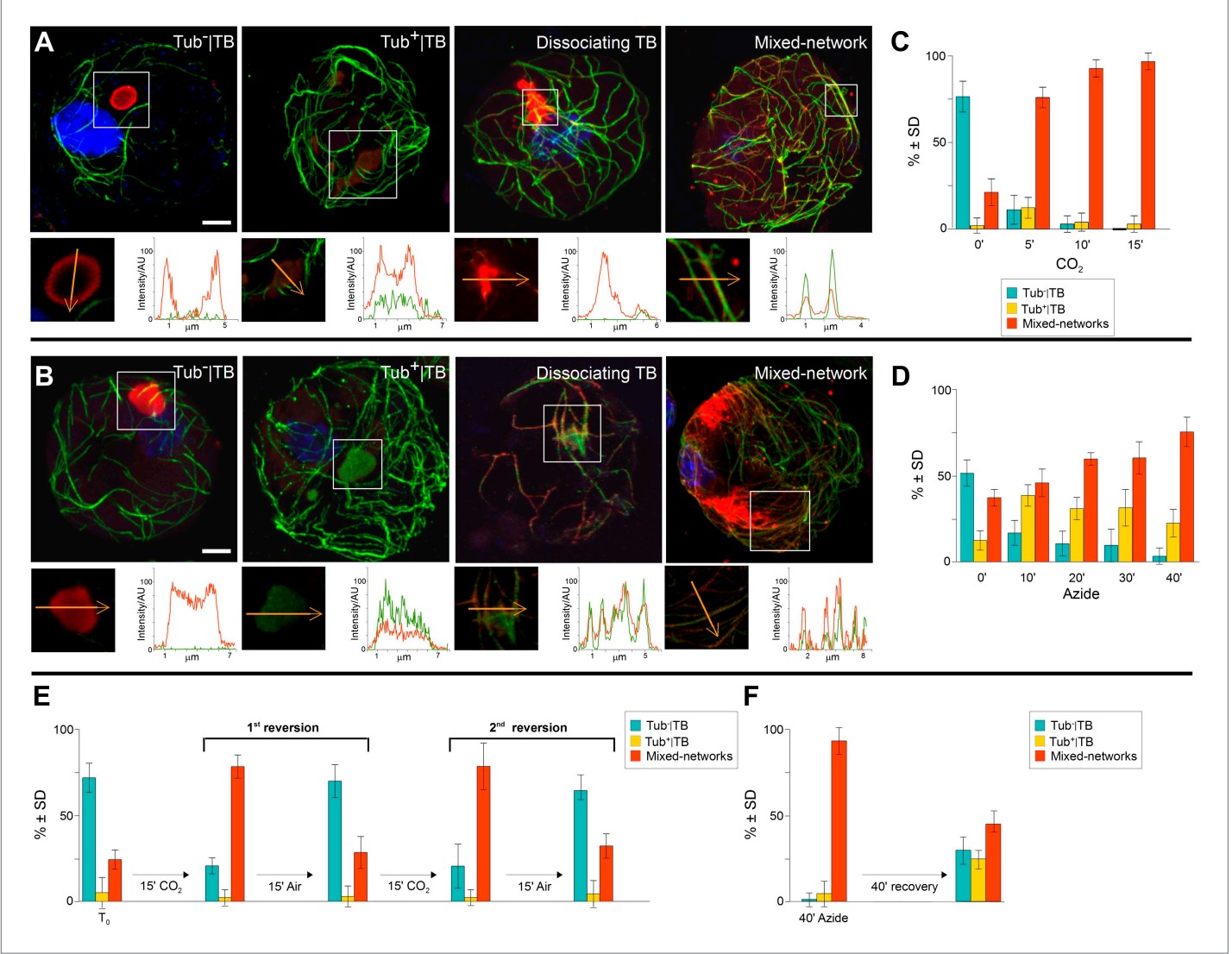

**Figure 5**. TB transformations have a precise temporal order and are reversible. (**A**–**B**) Kinetics of TB transformation. Protoplasts were treated with (**A**) $CO_2$ or (**B**) azide, and then processed for immunofluorescence against P2 (red) and α-tubulin (green); nuclei were counterstained with DAPI (blue). In both (**A**) and (**B**), untreated protoplasts display tubulin-less TBs, and the three subsequent images show representative treated protoplasts, respectively displaying a Tub+|TB, a disintegrating TB and mixed-networks. All images are confocal projections, with the exception of the dissociating TB after azide treatment, which is a single section; each inset shows a single optical section from the enclosed zone. The orange arrows show the line scans and the scanning direction used to create the profiles of P2 (red) and α-tubulin (green) labeling intensity (in arbitrary units, AU), which are displayed in the graphs to the right of the single sections. They reveal as in *Figure 2A–C*, that unstressed TBs display little to no tubulin label, whereas stressed TBs contain large amounts of tubulin in their centers. P2 colocalizes with microtubules in mixed-networks. The confocal stacks used to generate the image projections can be found in *Figure 5—source data 1 and 2*. (**C**–**D**) Quantification of TB kinetics. The histograms show the kinetics of $CO_2$- (**C**) and azide-triggered (**D**) TB transformation in protoplasts. Results from one out of three independent experiments are displayed. 1235 TBs (Tub−|TBs, Tub+|TBs, and mixed networks) were evaluated for the $CO_2$ experiments, and 1662 TBs were evaluated for the azide experiments. See *Figure 5—source data 3 and 4* for details. (**E**–**F**) Reversion of mixed-networks through two $CO_2$/air cycles (**E**), and after azide treatment (**F**). Infected protoplasts were treated with $CO_2$ or azide for the duration indicated. $CO_2$ was subsequently removed by ventilation of the suspension with air; azide was removed by resuspending the protoplasts in fresh medium. Shown are data from one of three independent experiments. For the three repetitions, a total of 1339 TB morphs were analyzed for $CO_2$ reversion, and 2262 TB morphs were analyzed for the azide reversion experiments. See *Figure 5—source data 5 and 6* for details. SD: standard deviation.

The following source data are available for figure 5:

**Source data 1.** Confocal single sections and acquisition parameters for *Figure 5A*

**Source data 2.** Confocal single sections and acquisition parameters for *Figure 5B*

*Figure 5. Continued on next page*

*Figure 5. Continued*

**Source data 3.** Source data for *Figure 5C*
**Source data 4.** Source data for *Figure 5D*
**Source data 5.** Source data for *Figure 5E*
**Source data 6.** Source data for *Figure 5F*

rate. We have previously shown that oryzalin diminishes transmission from infected protoplasts (*Martinière et al., 2011a*). Consistent with these results, we show here that oryzalin induces Tub$^+$|TBs within 15 min, but prevents mixed-network formation (*Figure 9F,G*).

To further confirm the role of TB changes in aphid-transmission of CaMV, we pursued an independent approach and examined a CaMV mutant impaired in transmission. This mutant, CaMV P2-TC, harbors a 7-amino acid insertion, including a tetracysteine tag (*Griffin et al., 1998*) at position 100 of the P2 protein. The mutant virus was fully infectious as compared to wild type virus; furthermore, the P2-TC protein as well as viral proteins P3, P4 and P6 accumulated to similar levels in infected plants (*Figure 10A*). However, in comparison to wild type TBs, this mutant induced TBs (TB-TCs) that seemed to be smaller, with a more regular rounded shape, and a more pronounced P2-rich cortex, as revealed by immunofluorescence (*Figure 10B*). Heat shock induced an influx of tubulin into TB-TCs, and photo-bleached GFP-tubulin contained in TB-TC was observed to be exchanged with cytoplasmic tubulin in FRAP experiments (*Figure 10C*), although the kinetics differed from tubulin replacement in wild type TBs (compare *Figure 10C* with *Figure 4B*). Thus the TB of the P2-TC mutant bears some similarity to wild type TB. Nevertheless, CaMV P2-TC was completely non-transmissible in plant-to-plant transmission experiments (*Figure 10D*). This could be due to a defect of TB-TC in undergoing correct transformation upon aphid puncture; an alternative is that this is due to a lack of interaction between the mutant P2-TC protein and either virus particles or aphid stylets. To distinguish between these possibilities, we first tested whether P2-TC protein can mediate the binding of virus particles to the aphid stylets. One way to test this is to allow aphids to feed on suspensions containing recombinant P2-TC, P3 and purified virus particles through membranes, before they are transferred to test plants for inoculation. Binding of transmissible complexes acquired by the aphids from the feeding solution is then scored by counting the number of successful transmission events, that is, the number of infected test plants. Although the transmission rates were significantly lower than that obtained with wild type P2, the P2-TC mutant protein was indeed active in such assays, retaining around 50% of the wild P2 activity (*Figure 10E*). We thus reasoned that defects in P2-TC binding to virus particles or stylets can only partially explain the complete failure in plant-to-plant transmission of the P2-TC mutant, and that a failure in TB-TC transformation may also be involved. To investigate this possibility, we allowed aphids to infest CaMV-P2TC-infected leaves for 15 min. The leaves were then processed for immunofluorescence against P2 and α-tubulin, and scored for the TB phenotype in cells found close to and farther away from the salivary sheaths. *Figure 10F,G* demonstrates that only standby TBs were detected, and not Tub$^+$|TBs or mixed-networks. This result is further evidence for a strong positive correlation between the appearance of mixed-networks (i.e., TB transformation) and successful aphid transmission, and presents direct biological evidence in support of our transmission hypothesis.

## Discussion

Combined, our results establish that the TB is a dynamic structure, which in the presence of an aphid vector can react immediately to promote CaMV transmission (*Figure 11*). Transmission is controlled by the different TB morphs. The tubulin-less TB found in normal 'unstressed' cells has been well-described over the past several decades (*Shalla et al., 1980*; *Rodriguez et al., 1987*; *Espinoza et al., 1991*; *Blanc et al., 1993b*; *Drucker et al., 2002*), and here functions as a standby TB that 'anticipates' the vector arrival. Then, when aphids land on the plant and insert their stylets into tissues, CaMV uses the plant's response during stylet entry to its advantage at a very early stage of the plant–aphid interaction. This results in TBs that undergo dramatic, short-lived changes leading to the temporary redistribution

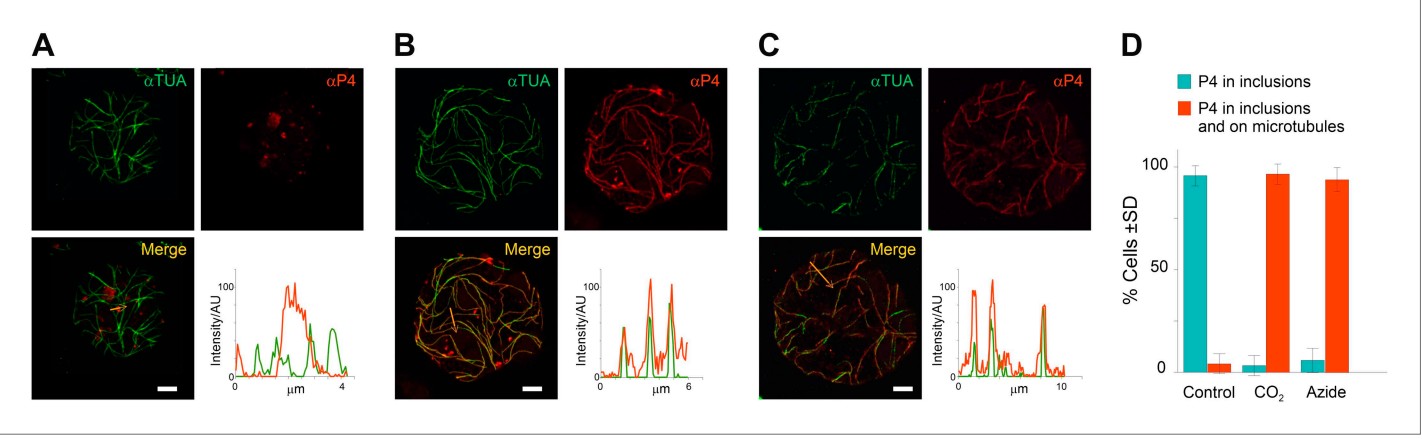

**Figure 6**. TB transformation mobilizes virus particles onto microtubules. (**A–C**) Viral capsid protein P4 colocalizes with mixed-networks. Protoplasts were either left unstressed (**A**), incubated with $CO_2$ for 15 min (**B**) or treated with azide for 40 min (**C**), and then fixed and labeled to detect capsid protein P4 (red) and α-tubulin (αTUA, green). The split channel representations and merges as well as the P4 and α-tubulin profiles obtained by scanning the lines indicated by the orange arrows show that the two stress treatments induced relocalization of capsid protein P4 from inclusions onto microtubules. As in **Figure 2A–C**, the intensity of the P4 and α-tubulin label is indicated in arbitrary units (AU) because different acquisition settings were used to record the images. (**A**) is a confocal projection, (**B–C**) are confocal single sections. Refer to **Figure 6—source data 1–3** for image details. (**D**) Quantification of the effect of azide and $CO_2$ on the localization of P4. Cells were treated as indicated, processed for immunofluorescence against P4 and α-tubulin and scored for the presence of P4 in inclusions only, or in inclusions and on microtubules. The histogram shows that almost all cells display P4 networks that colocalize with microtubules after stress treatment. Data are from one of three independent experiments, in which a total of 524 cells were analyzed. Refer to **Figure 6—source data 4** for details. Scale bars: 5 μm. SD: standard deviation.

The following source data are available for figure 6:

**Source data 1.** Confocal projection and acquisition parameters for **Figure 6A**.

**Source data 2.** Confocal single section and acquisition parameters for **Figure 6B**.

**Source data 3.** Confocal single section and acquisition parameters for **Figure 6C**.

**Source data 4.** Source data for **Figure 6D**.

of P2 and virus onto microtubules. These hitherto overlooked TB alterations result in a reversible TB 'activation' that optimizes virus acquisition. The activation or increased transmission efficiency is probably due to the mixed-networks distributing P2 and virus homogeneously on microtubules throughout the cell periphery. This could in turn facilitate virus acquisition: in this new configuration, P2 and the virus are more accessible to the vector during its random punctures, as compared to the remote localization of P2 in isolated TBs. This hypothesis is supported by the observation that inhibition of mixed-network formation, either in the CaMV-P2TC mutant or pharmacologically by oryzalin, resulted in decreased transmission. The converse situation also applies, as artificial induction of mixed-networks by $CO_2$ and azide were correlated strongly with increased transmission rates. These results additionally indicate that the TB reaction is required for transmission.

One enduring question that deserves further attention is precisely where the virus particles come from that are recruited onto the microtubules. Whereas the origin of microtubule-associated P2 is clearly the TB (since the TB is the only source of P2), the origin of the virus particles aligning on the microtubules is less clear. They could derive either from the TB, which contains some virus particles, or from the many virus factories dispersed throughout the cytoplasm (**Espinoza et al., 1991**; **Drucker et al., 2002**). We were also intrigued to observe that only certain stresses—notably aphid feeding activity, wounding, azide and $CO_2$—could trigger TB transformation. This reveals a certain level of specificity in TB activation, although it is more broad than the classical pattern recognition receptor-mediated defense responses of plants against pathogens (including insects) that are often species or even isolate-specific (**Hogenhout and Bos, 2011**). This broad specificity is not surprising,

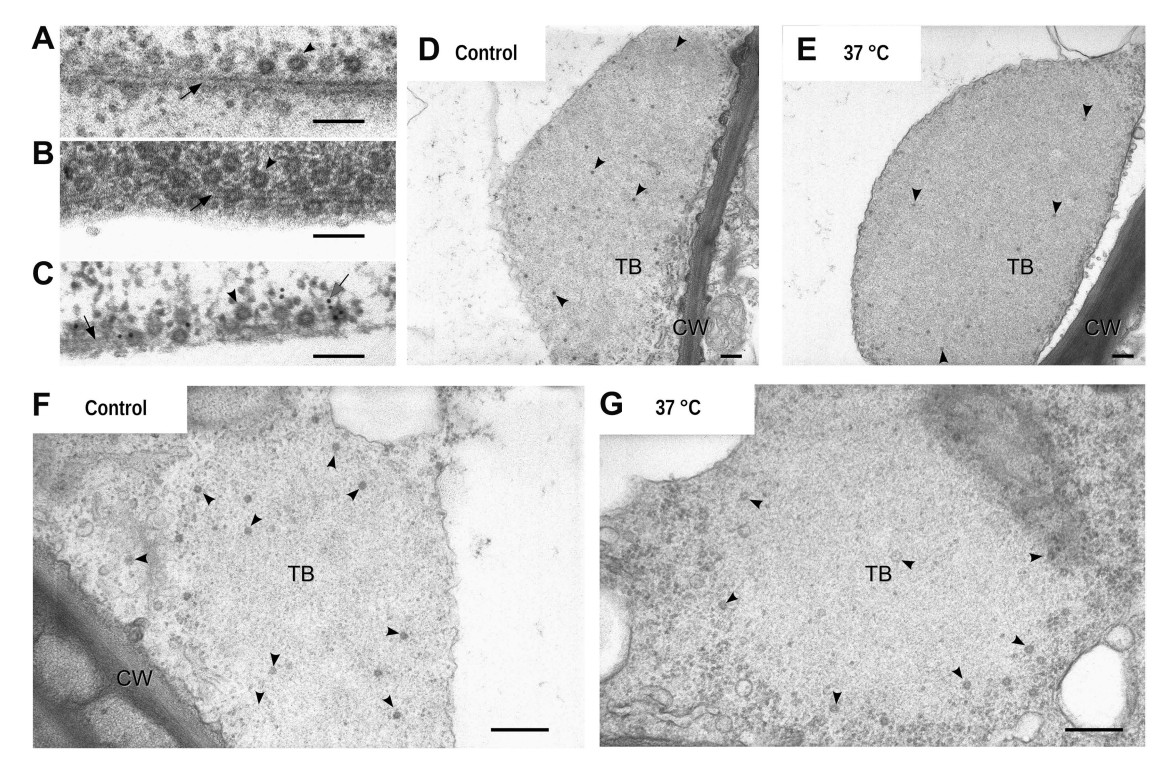

**Figure 7**. Electron microscopy of the different TB morphs. (**A–C**) Mixed-networks display virus particles on microtubules. The images show typical spherical CaMV virus particles (arrowheads) that decorate microtubules (black arrows) in cortical regions of (**A**) a $CO_2$-treated or (**B**) an azide-treated protoplast. (**C**) Positive immunogold labeling against P2 (the gray arrow points to an exemplary nanogold particle) identifies the virus-decorated microtubules as mixed-networks, in which all components of the CaMV transmissible complex are present. (**D–G**) TBs in unstressed (**D**, **F**) and heat-shocked (**E**, **G**) tissue display the same TB phenotype. Infected Arabidopsis *TUA6-GFP* leaves were exposed for 1 h at 37°C. The presence of tubulin in TBs was then verified by fluorescence microscopy and the same leaf samples were processed for transmission electron microscopy. The arrowheads point to virus particles. CW: cell wall. For scale bars, (**A–C**): 100 nm; (**E–G**): 250 nm.

since CaMV is transmitted by at least 30 different aphid species. Importantly, this suggests that the plant responses against aphids (that are probably exploited by CaMV for the TB reaction) are triggered by an elicitor common to all aphids. Whether this elicitor triggers an innate plant immunity pathway or a separate perception/reaction cascade remains an open question.

The significance of this remarkable phenomenon described here extends beyond CaMV transmission to broader fields of research. First, this opens up a fascinating new direction within virology, to explore whether other viruses form transmission morphs in response to vector-sensing by the host. Second, the transient accumulation of tubulin in TBs, followed by redistribution of TB contents on microtubules uncovers unforeseen capacities of tubulin/microtubule dynamics and raises further questions pertinent to cell biology. For example: How can apparently soluble tubulin concentrate in the TB and what is its function there? Does this serve as the source for the mixed-networks? Most strikingly, our work reveals that a virus can detect external stresses (probably by using its host's perception system) and respond in a way that is somewhat independent of the host's response. We propose naming this phenomenon: 'virus perceptive behavior'. This concept is nicely illustrated by three compelling observations made in this study. First, TB transformation occurs while the host plant is still in the process of transducing the triggering signal. This shows that the virus appropriates the host's perception machinery itself, rather than relying upon downstream reactions that take tens of minutes (or hours) to manifest and establish local and systemic defense responses (*Kuśnierczyk et al., 2008*; *de Vos and Jander, 2010*). Second, after transformation of the TB into mixed-networks, the fate of CaMV appears disconnected from the final host response. Indeed, within the time frame required for the host plant to respond to an aphid attack, mixed-networks have already served as a robust virus

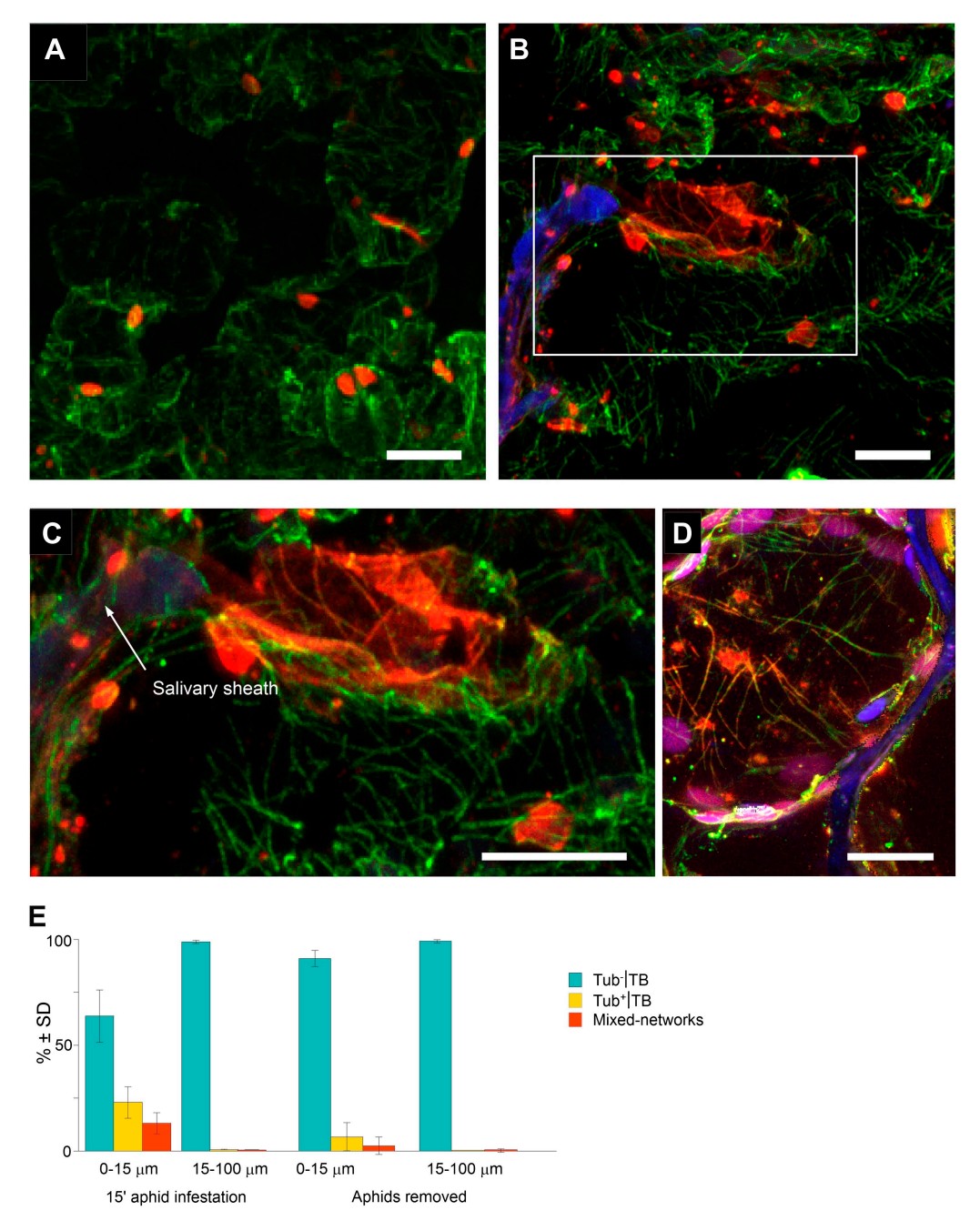

**Figure 8**. Mixed-networks appear in tissue zones pierced by aphid stylets. Unstressed CaMV-infected leaves (**A**) or leaves infested by aphids for 15 min (**B**–**D**) were analyzed by immunofluorescence microscopy. (**A**) Cells in leaf regions that were not foraged by aphids display standby Tub⁻|TBs, as shown by confocal projections of tissue sections labeled against P2 (red) and α-tubulin (green). The optical single sections used for this projection are deposited in ***Figure 8—source data 1***. (**B**) In contrast, a cell close to a salivary sheath (blue autofluorescence, digitally enhanced) displays mixed-networks, in aphid-infested tissue. An enlargement of the zone enclosed in (**B**) is shown in (**C**). (**B**–**C**) show confocal projections, please refer to ***Figure 8—source data 2*** for the corresponding image stack. (**D**) Immunofluorescence microscopy against capsid protein P4 (red) and α-tubulin (green) shows that virus particles also localize to mixed-networks in cells close to salivary sheaths (blue, digitally enhanced). Chloroplast autofluorescence appears in magenta. The confocal single sections used to produce this projection can be found in ***Figure 8—source data 3***. (**E**) Aphids trigger TB transformation, and this transformation is reversible. Aphids were placed for 15 min on infected leaves. Following this, the leaves were fixed immediately

*Figure 8. Continued on next page*

*Figure 8. Continued*

and processed for immunofluorescence (15-min aphid infestation), or the aphids were removed and the leaves were processed 2 h later (Aphids removed). The TB phenotype (standby Tub⁻|TBs, Tub⁺|TBs and mixed-networks) was scored next to salivary sheaths (0–15 µm) and in surrounding tissue (15–100 µm). Tub⁺|TBs and mixed-networks were predominantly observed close to salivary sheaths in freshly aphid-infested tissue. The effect was highly significant (p<0.0001, GLM, df = 1, $\chi^2$ = 194.59, n = 3). Tub⁺|TBs and mixed-networks reverted back to 'stand-by' Tub⁻|TBs 2 h after aphid removal, indicating that TB activation is reversible. This effect was also highly significant (p<0.0001, GLM, df = 1, $\chi^2$ = 17.98, n = 3). SD in (**E**): standard deviation from three independent experiments. A total of 969 TBs surrounding 42 sheaths were counted from freshly aphid-infested tissue, and 194 TBs surrounding eight sheaths were counted in the 'aphids removed' experiments. Original data can be found in **Figure 8—source data 4**.

The following source data are available for figure 8:

**Source data 1.** Confocal single sections and acquisition parameters for **Figure 8A**.
**Source data 2.** Confocal single sections and acquisition parameters for **Figure 8B**.
**Source data 3.** Confocal single sections and acquisition parameters for **Figure 8D**.
**Source data 4.** Source data for **Figure 8E**.

source for aphids, and have long since reverted back to standby TBs. Third, the response mechanisms themselves are also entirely different from the reported plant physiological responses to aphid attack. These include callose deposition near the salivary sheaths and within sieve tubes (*Villada et al., 2009*), changes in gene expression patterns (*Kuśnierczyk et al., 2008*), altered emission of volatile compounds (*de Vos and Jander, 2010*); and the initiation of salicylic, abscisic, and jasmonic acid systemic defense pathways (*Giovanini et al., 2007*; *Kuśnierczyk et al., 2008*; *de Vos and Jander, 2009*). In contrast, the TB response seems to be restricted to a CaMV-specific and immediate diversion of tubulin/microtubules (plus putative unknown associated partners) for virus transmission, in a manner unlike anything described before.

Whether such viral perceptive behaviors play a role in the vector-transmission of other viruses is entirely unknown, and will thus be a question of great priority in the field of research on virus transmission. Viruses tightly regulate all the different steps of their life cycle, from intracellular replication and short- and long-distance intra-host movement, to inter-host spread. In this sense, the ability to specifically trigger the 'transmission-mode' at the right time and the right place seems like a valuable adaptation for avoiding the deleterious interference between these various functions.

On a more broad scope, our results highlight many unexpected research horizons to explore in the biology of these fascinating pathogens. The possible instances in which viruses could react directly to cues from the host environment, the diversity of sensorial pathways that could be exploited in both animal and plant hosts, and the number of key life cycle steps that could be optimized accordingly all inspire questions that will shape future research directions in this field. Finally, aside from being an academic challenge, this phenomenon also represents a potential Achilles heel in viral transmission that could lead to novel virus control strategies.

## Materials and methods

### Plants, viruses and inoculation

Turnip plants (*Brassica rapa* cv. 'Just Right') and transgenic *Arabidopsis thaliana* Col0 plants with a *gl1* marker expressing GFP-TUA6 under control of the 35S promoter (*Ueda et al., 1999*) were alternatively used as CaMV hosts, depending on the experiment. Two-week-old plants were mechanically inoculated with wild-type CaMV strain Cabb B-JI (*Delseny and Hull, 1983*) or Cabb B-JI ΔP2 as described in *Martinière et al. (2009)*, and processed as indicated at 14 days post infection (dpi). In order to obtain the mutant virus Cabb B-JI P2-TC (referred to as P2-TC in the text), the oligonucleotides 5'-TCGAGTTGCTGTCCAGGATGTTGC-3' and 5'-TCGAGCAACATCCTGGACAGCAAC-3' were

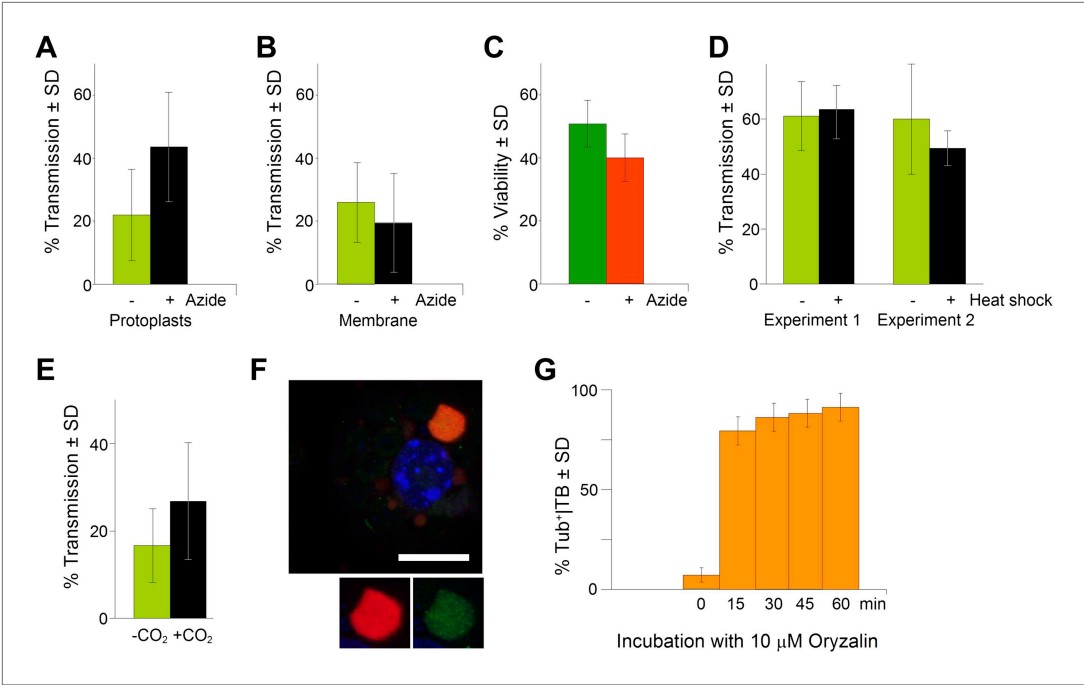

**Figure 9**. TB transformation correlates with enhanced transmission efficiency. (**A**) Azide enhances transmission from protoplasts. Aphids were allowed to acquire CaMV from infected protoplasts that displayed mixed-networks induced by azide. They were then transferred to healthy test plants for inoculation, and infected plants were counted 3 weeks later. The difference in transmission was highly significant (p<0.0001, hierarchical GLM model, **Table 3** and **Figure 9—source data 1**). (**B**) Azide does not affect aphid behavior. To rule out an unwanted effect of azide on aphid viability and behavior, aphids were membrane-fed solutions containing purified virus particles, recombinant P2 and P3, in the presence or absence of azide, and then transferred to healthy test plants for inoculation. Transmission rates were determined 3 weeks later by scoring infected plants. Data from one experiment are shown, using three different virus preparations as a virus source for each condition. See **Figure 9—source data 2** for details. (**C**) Azide does not affect protoplast viability. Protoplasts were incubated for 1 h (the duration of a transmission test) in the presence or absence of 0.02% azide, and then protoplast viability was determined with the FDA test (**Widholm, 1972**). Data from one out of two experiments are shown. The difference in viability was insignificant in this experiment (p=0.0658, n = 6, Mann–Whitney test) and also in the second experiment. See **Figure 9—source data 3** for all data. (**D**) Heat shock does not enhance CaMV transmission. Leaves from *GFP-TUA6* Arabidopsis either received heat shock (+) or did not (−). The presence of Tub[+]|TBs was verified by fluorescence microscopy and the leaves were then used in aphid transmission assays. No significant difference in transmission was observed in either of two independent experiments (p=0.73 and p=0.08, respectively, hierarchical GLM model, see **Table 4** and **Figure 9—source data 4**). (**E**) $CO_2$ enhances CaMV transmission. Leaves with mixed-networks induced by $CO_2$ were used in plant-to-plant aphid transmission experiments. $CO_2$-treated leaves performed significantly better in transmission tests than controls (p=0.0025, hierarchical GLM model, see **Table 5** and **Figure 9—source data 5**). (**F**) Oryzalin induces Tub[+]|TBs. Immunofluorescence of oryzalin-treated protoplasts shows that α-tubulin (green) accumulates with P2 (red) in TBs. The nucleus is stained with DAPI (blue). The image is a confocal projection. The insets show a separate channel presentation of a representative optical single section of the TB, for details refer to the image stack in **Figure 9—source data 6** that was used for this projection. Scale bar = 10 μm. (**G**) Kinetics of Tub[+]|TB formation in protoplasts that were treated with oryzalin for the duration indicated. Most TBs transformed to the Tub[+]-state within 15 min. Mixed-networks were not observed and thus are not indicated in the histogram. Data is from one of three independent experiments, where a total of 1556 TBs were analyzed. See **Figure 9—source data 7** for details. SD: standard deviation.

The following source data are available for figure 9:

**Source data 1.** Source data for **Figure 9A**.

**Source data 2.** Source data for **Figure 9B**.

*Figure 9. Continued on next page*

*Figure 9. Continued*

**Source data 3.** Source data for *Figure 9C*.

**Source data 4.** Source data for *Figure 9D*.

**Source data 5.** Source data for *Figure 9E*.

**Source data 6.** Confocal single sections and acquisition parameters for *Figure 9F*.

**Source data 7.** Source data for *Figure 9G*.

**Table 3.** Statistical analysis of transmission experiments using azide-treated protoplasts as virus source

| Experiment | n | Transmission frequency | LCI | UCI |
|---|---|---|---|---|
| Global | | | | |
| −Azide | 16 | 0.24 | 0.18 | 0.30 |
| +Azide | 16 | 0.40 | 0.33 | 0.47 |
| Experiment 1 | | | | |
| −Azide | 3 | 0.32 | 0.22 | 0.43 |
| +Azide | 3 | 0.38 | 0.27 | 0.50 |
| Experiment 2 | | | | |
| −Azide | 3 | 0.44 | 0.32 | 0.55 |
| +Azide | 3 | 0.66 | 0.54 | 0.76 |
| Experiment 3 | | | | |
| −Azide | 3 | 0.19 | 0.11 | 0.29 |
| +Azide | 3 | 0.41 | 0.30 | 0.52 |
| Experiment 4 | | | | |
| −Azide | 3 | 0.33 | 0.22 | 0.44 |
| +Azide | 3 | 0.46 | 0.35 | 0.58 |
| Experiment 5 | | | | |
| −Azide | 4 | 0.10 | 0.03 | 0.19 |
| +Azide | 4 | 0.38 | 0.29 | 0.48 |

We measured the transmission rate by aphids for each condition of treatment (−azide and +azide) for the five experiments. Since a non-significant interaction between experiments and treatments was found (GLM, df = 4, $\chi^2$ = 6.69, p=0.15), the data was pooled in the line named 'global'. Azide induced a highly significant increase of the transmission rate compared to the control without azide (hierarchical GLM model using Firth's penalized likelihood, df = 1, $\chi^2$ = 35.29, p<0.0001) with a transmission rate of 23.7% (95% CI: 17.6–29.7%) for control and 39.9% (33.2–46.6%) for azide treatment.

CI: confidence interval; n: number of repetitions per experiment; LCI, UCI: lower and upper limits of confidence intervals, respectively.

annealed. This created *Xho*I-compatible restriction sites at the two extremities of the then double-stranded oligonucleotide that were used for insertion into the unique *Xho*I site in the Cabb B-JI genome cloned into the pCa24 plasmid (*Delseny and Hull, 1983*). Positive clones were identified by PCR and verified by sequencing. They contained a seven-amino-acid insertion at amino acid position 100 of the P2 open reading frame, coding for a tetracysteine tag (CCPGCC [*Griffin et al., 1998*]) as well as an additional serine.

## Aphids

A non-viruliferous clonal *Myzus persicae* population was reared under controlled conditions (22/18°C day/night with a photoperiod of 14/10 h day/night) on eggplant and cultivated by G. Labonne (INRA, Montpellier). The population was started from a single virginiparous female.

## Recombinant P2-TC

To produce recombinant P2-TC using the *Sf9*/baculovirus system, the tetracysteine sequence and an additional serine were introduced into the unique *Xho*I site in the P2 coding region of plasmid p119-P2, using the same strategy as for cloning CaMV-P2TC described above. Recombinant baculovirus was obtained by homologous recombination as described in *Blanc et al. (1993b)*. Infected *Sf9* cells were harvested 48 h after inoculation and total cell extracts were prepared in SES buffer and stored at −20°C until use.

## Isolation of protoplasts

Protoplasts were prepared from healthy or infected (14 dpi) leaves of turnip plants as described in *Martinière et al. (2009)*. Briefly, leaves were sterilized by submerging them in 20-fold diluted Domestos solution (http://www.unilever.com) for 3 min. The leaves were then washed three times with water, prior to overnight incubation in

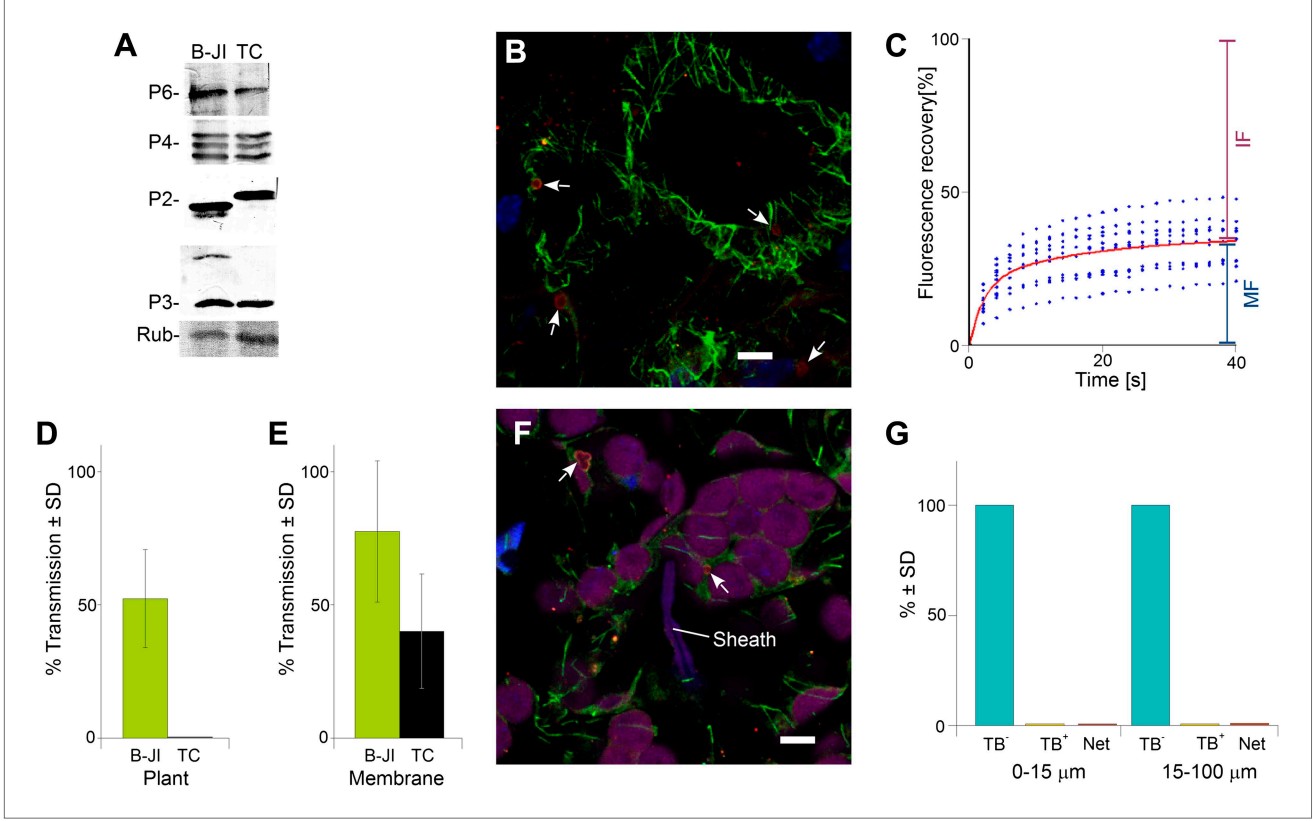

**Figure 10**. The P2-TC mutant of CaMV is inactive in plant-to-plant transmission. (**A**) Accumulation of viral proteins in CaMV P2-TC-infected plants. Western blot analysis of total leaf extracts shows that virus factory protein P6, the three forms of capsid protein P4, as well as P2 and P3 accumulate to similar levels in plants infected with wild type CaMV (B-JI) or the P2-TC mutant (TC). Rub = Rubisco loading control stained with Ponceau Red. (**B**) CaMV P2-TC-infected plants display TBs. Confocal projection of infected leaf sections labeled for P2 (red) and α-tubulin (green) shows that the CaMV mutant P2-TC forms TBs (arrows) that are smaller and more regular than wild type TBs. Nuclei are counterstained with DAPI (blue). The optical single sections used for the projection are presented in *Figure 10—source data 1*. (**C**) Tubulin turnover in P2-TC mutant TBs. Arabidopsis plants constitutively expressing GFP-tubulin were infected with the CaMV P2-TC mutant. Leaf epidermis was screened by fluorescence microscopy for GFP-tubulin inclusions that were identified as TBs based on their typical shape. The GFP-tubulin was photobleached in these TBs, and the recovery of the GFP-fluorescence (due to replacement of the photobleached GFP-tubulin by fresh tubulin) was recorded in FRAP experiments. The graph shows recovery kinetics from t = 0 s (time point of photobleach) onwards. The fluorescence levels were normalized (100% = fluorescence before bleaching, 0% = fluorescence just after bleaching). The red trend line was calculated from nine experiments. Data points from the nine experiments are indicated as blue dots. These results indicate that tubulin cycles between the cytoplasm and mutant TBs, albeit with different kinetics than for wild type TBs (compare with *Figure 4C*). Compared to wild type TBs, the $t_{(1/2)}$ for fluorescence recovery was significantly slower (p<0.0001, t-test with n = 21 for wild type TBs and n = 23 for TC-TBs) and the proportion of the mobile fraction was significantly higher in P2-TC TBs (p=0.0001, t-test with n = 21 for wild type TBs and n = 23 for TC-TBs). Refer to *Figure 10—source data 2* and *Figure 4—source data 1* (wild type TBs) for data sets. (**D**) The mutant P2-TC does not support plant-to-plant transmission. Aphids were placed for 15 min on CaMV wild type-infected (B-JI) or P2-TC-infected (TC) leaves and then transferred to healthy test plants for inoculation. Infected plants were scored 3 weeks later. Pooled data are shown from two independent experiments using 12 different leaves for each condition. No statistical analysis was performed, as the effect of the P2-TC mutant on plant-to-plant transmission was total (no transmission from P2-TC-infected plants was observed). See *Figure 10—source data 3* for the data sets. (**E**) The P2-TC protein itself is active in transmission. Recombinant wild type P2 (B-JI) or mutant P2-TC (TC) were mixed together with recombinant P3 protein and purified CaMV particles. Aphids were allowed to feed on the suspensions across membranes for 15 min and were then transferred to healthy test plants for inoculation. Infected plants were counted 3 weeks later. The histogram shows that P2-TC supported aphid transmission of CaMV under these conditions, although this was significantly reduced as compared to the wild type P2 (p=0.02, n = 8 from two independent experiments, Mann–Whitney test). See *Figure 10—source data 4* for data. (**F**) Aphid stylet activity does not trigger TB transformation in CaMV P2-TC-infected leaves. Aphids were allowed to feed on CaMV-P2-TC-infected leaves for 15 min. The tissue was then processed for immunofluorescence against P2 (red) and α-tubulin (green); nuclei were stained with DAPI (blue). The confocal projection in (**F**) indicates that cells in contact with a salivary sheath (Sheath) display tubulin-less TBs (arrows). Chloroplasts are displayed in magenta to better distinguish the cells. Please see *Figure 10—source data 5* for the confocal single sections used to create the projection. (**G**) Quantitative analysis of the TB forms of CaMV-P2-TC in aphid-infested tissue reveals the absence of Tub+|TBs and mixed-networks, both close to salivary sheaths (0–15 µm) and farther away (15–100 µm). Data shown are from three independent experiments where a total of 510 TBs were analyzed (see *Figure 10—source data 6* for details). As the effect was total, no statistical analysis was performed. SD: standard deviation.

*Figure 10. Continued on next page*

*Figure 10. Continued*

The following source data are available for figure 10:

**Source data 1.** Confocal single sections and acquisition parameters for *Figure 10B*.

**Source data 2.** Source data for *Figure 10C*.

**Source data 3.** Source data for *Figure 10D*.

**Source data 4.** Source data for *Figure 10E*.

**Source data 5.** Confocal single sections and acquisition parameters for *Figure 10F*.

**Source data 6.** Source data for *Figure 10G*.

**Table 4.** Statistical analysis of transmission experiments using heat-shocked leaves as virus source

| Experiment | n | Transmission frequency | LCI | UCI |
|---|---|---|---|---|
| Experiment 1 | | | | |
| Control | 6 | 0.62 | 0.54 | 0.70 |
| 37°C | 6 | 0.64 | 0.40 | 0.72 |
| Experiment 2 | | | | |
| Control | 6 | 0.60 | 0.51 | 0.68 |
| 37°C | 6 | 0.49 | 0.40 | 0.57 |

Data for experiments 1 and 2 were analyzed independently, as heat-shock slightly increased and decreased transmission in these experiments, respectively. The heat shock treatment (90 min at 37°C) induced no significant difference in transmission rate (compared to controls) for experiment 1 (hierarchical GLM, df = 1, $\chi^2$ = 0.12, p=0.73), with 62% (95% CI: 53.6–70%) for control and 64% (55.8–71.7%) for heat-shock treatment, or for experiment 2 (hierarchical GLM, df = 1, $\chi^2$ = 3.12, p=0.08), with 59.5% for control (50.6–67.9%) and 48.6 % (40.4–56.9%) for heat-shock treatment.
n: number of repetitions per experiment; LCI, UCI: lower and upper limits of confidence intervals, respectively.

protoplast medium M (0.5 M mannitol, 1 mM $CaCl_2$, 10 mM MES, pH 5.8) containing freshly added 0.5% cellulose 'Onozuka' R10 and 0.05% macerozyme R10 (both enzymes obtained from Yakult, http://www.yakult.co.jp/ypi/). Protoplasts were separated from undigested tissue by filtration through Miracloth (http://www.merckmillipore.com), and washed three times with buffer M by centrifugation at 80×*g* for 5 min in a swing-out rotor. Prior to treatments, protoplasts were maintained fourfold diluted in buffer M at room temperature with slow agitation (5 rpm) for 2 h.

## Drug and stress treatments

The different drug treatments with respective experimental times were: 10 μM oryzalin (1 h), 0.02% azide (40 min), or $CO_2$ atmosphere (15 min). Azide (100× concentration) or oryzalin (1000× concentration) stock solutions were added to water or DMSO, respectively. Pure solvent was used as a control. For $CO_2$ treatment, leaves or protoplasts were placed in a plastic box filled with $CO_2$ that was generated by sublimation of dry ice in water, contained in a small beaker in the box. We visually confirmed the displacement of the water cloud initially created by the subliming $CO_2$ to assure that the heavier $CO_2$ had replaced the air (no more water vapor visible); only then was the plant material placed in the box. We also verified that the dry ice did not lower the temperature of the atmosphere in the box. For TB reversion, protoplasts were cycled every 15 min between the plastic box and standard bench-top conditions for the $CO_2$ treatment or the azide was removed by replacing the protoplast medium with fresh medium after centrifugation of the protoplasts for 5 min at 80×*g* in a swing-out rotor. Protoplast viability was verified by the fluorescein diacetate test (*Widholm, 1972*). For heat shock treatment, protoplasts or plants were placed in an incubator at 37°C. To inflict wounding, leaves were cut with a new razor blade.

## Histology

Leaf segments (5- to 7-mm-long) of turnip or Arabidopsis were fixed in 1% glutaraldehyde prepared in stabilizing buffer (50 mM HEPES, pH 8). The tissue was then embedded in either Steedman's wax, as described in *Vitha et al. (2000)*, or in 5% agarose. For the Steedman's wax method, the leaf

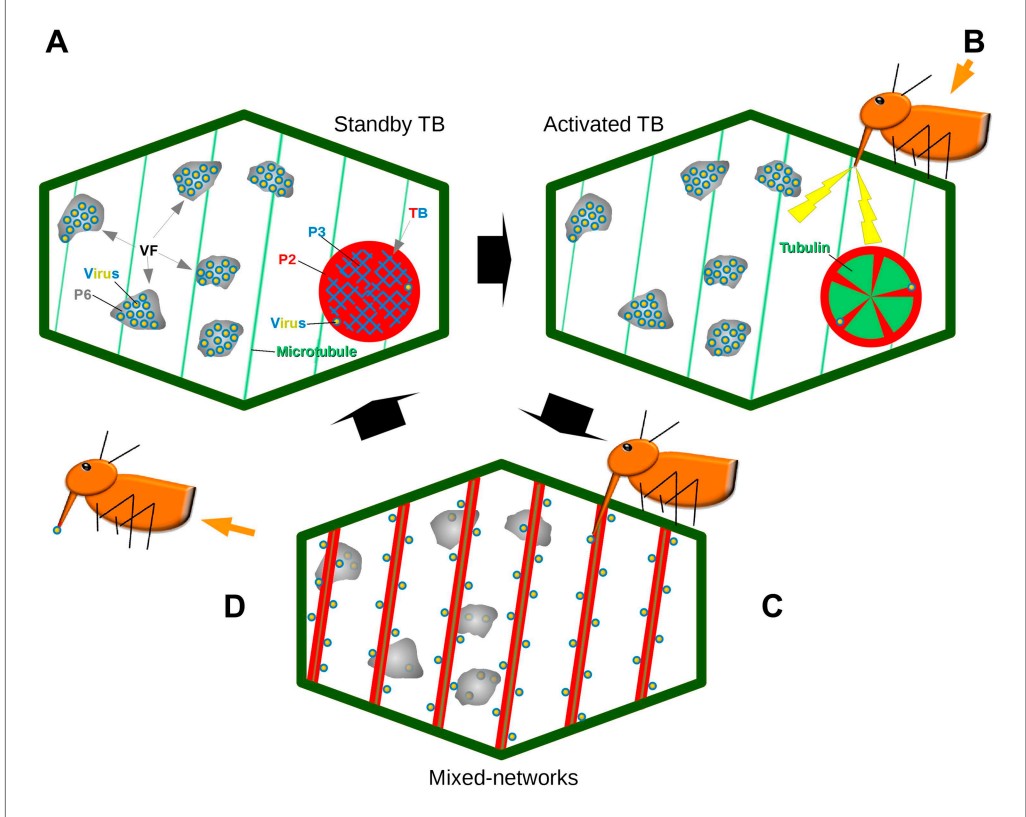

**Figure 11**. Model of CaMV acquisition. (**A**) In an infected cell in the 'standby' state, there are numerous virus factories (VF) containing most of the replicated virus particles (yellow-blue circles), enclosed within a matrix of viral protein P6 (grey). This is accompanied in the cell by a mostly single transmission body (TB), composed of a matrix containing all of the cell's P2 (red), co-aggregated with P3 (blue) and some virus particles. Microtubules are represented in green. (**B**) An aphid landing on an infected plant inserts its stylets into a cell to test the plant. This causes a mechanical stress (stylet movement) and/or a chemical stress (e.g., elicited by saliva components). This stress, symbolized by the yellow flashes, is immediately perceived by the plant and can induce subsequent defense responses. The initial aphid recognition signal is transduced simultaneously in a TB response, characterized by an influx of tubulin (green) into the TB. (**C**) In a second step, the TB disintegrates rapidly (within seconds), and all the P2 as well as some virus particles relocalize on the cortical microtubules as mixed-networks. Whether or not the virus particles originate at the VFs, as presented here, is unknown. Transmissible P2-virus complexes are now homogeneously distributed throughout the cell periphery, which significantly increases the chances of successful binding of P2 and virus to the stylets and thus transmission. (**D**) After departure of the aphid vector (here loaded with P2 and virus), a new TB is reformed from the mixed-networks and is ready for another round of transmission.

segments were rinsed twice for 10 min each with 50 mM HEPES pH 8, followed by two 10-min washes with PBS. After dehydration through an ethanol series, the samples were infiltrated at 40°C with Steedman's wax by using a graded ethanol/wax series. Finally, the segments were embedded in pure Steedman's wax. After polymerization of blocks at room temperature, a microtome was used to cut slices with a 14-μm thickness. For the agarose method, the leaf segments were rinsed twice for 5 min in PBS and then embedded in 5% low melting temperature agarose. Finally, 50 μm sections were cut with a vibratome.

## Electron microscopy

Samples for electron microscopy and immunoelectron microscopy were processed as described in *Drucker et al. (2002)*. For transmission electron microscopy, infected leaves or agarose-embedded protoplasts were fixed with 4% glutaraldehyde, postfixed with 2% OsO$_4$, and embedded in Epon resin (http://www.emsdiasum.com). For immunoelectron microscopy, protoplasts were fixed with

**Table 5.** Statistical analysis of transmission experiments using $CO_2$-treated leaves as virus source

| Experiment | n | Transmission frequency | LCI | UCI |
|---|---|---|---|---|
| Global | | | | |
| Control | 16 | 0.13 | 0.09 | 0.17 |
| $CO_2$ | 16 | 0.22 | 0.18 | 0.27 |
| Experiment 1 | | | | |
| Control | 4 | 0.11 | 0.05 | 0.19 |
| $CO_2$ | 4 | 0.15 | 0.07 | 0.25 |
| Experiment 2 | | | | |
| Control | 6 | 0.14 | 0.08 | 0.21 |
| $CO_2$ | 6 | 0.21 | 0.14 | 0.28 |
| Experiment 3 | | | | |
| Control | 6 | 0.13 | 0.07 | 0.20 |
| $CO_2$ | 6 | 0.30 | 0.22 | 0.39 |

We measured the transmission rate by aphids for each treatment condition (control or $CO_2$) for the three experiments. As we found a non-significant interaction between experiments and treatments (GLM, df = 2, $\chi^2$ = 2.92, p=0.23), the three experiments were pooled for analysis (line named 'global'). $CO_2$ induced a significant increase in the transmission rate compared to the controls without $CO_2$ (hierarchical GLM model, df = 1, $\chi^2$ = 9.12, p=0.0025), with 13.0% (95% CI: 9.4–17.0%) for control in ambient atmosphere and 22.4% (17.8–27.2%) for $CO_2$-treatments.

n: number of repetitions per experiment; LCI, UCI: lower and upper limits of confidence intervals, respectively.

0.5% glutaraldehyde and 2% paraformaldehyde and embedded in LR Gold resin (http://www.emsdiasum.com). All primary antisera and secondary antibodies were used at 1:25 dilution. The grids were observed in a Jeol JEM 100CX II electron microscope (http://www.jeol.com) operated at 60–80 kV.

## Antisera
The following antibodies or antisera were used: rabbit anti-P2 (*Blanc et al., 1993b*), anti-P3 (*Drucker et al., 2002*), anti-P6 (*Khelifa et al., 2007*), monoclonal mouse anti-α-tubulin DM1A (http://www.sigmaaldrich.com; *Blose et al., 1984*), and rabbit anti-P4 (http://plant.neogeneurope.com). For secondary antibodies, we used Alexa 488 and Alexa 594 conjugates (http://www.lifetechnologies.com) or 10 nm colloidal gold conjugates (http://www.bbigold.com).

## Western blotting
Plant tissue was ground in liquid nitrogen, and the powder was resuspended in 2× Laemmli buffer (*Laemmli, 1970*) and boiled for 5 min. After brief centrifugation in a tabletop centrifuge (5 min at 16,000×*g*), aliquots were separated by SDS/PAGE using 12% gels. Proteins were transferred onto nitrocellulose membranes and antigens were revealed by the NBT-BCIP reaction as described in *Drucker et al. (2002)*.

## Immunofluorescence of protoplasts
After treatments, protoplasts were fixed for 20 min at room temperature with 1% glutaraldehyde in 0.5 M mannitol and 50 mM HEPES pH 8, and washed with TS buffer (50 mM Tris, 150 mM NaCl, pH 7.4). Protoplasts were immobilized on polylysine-coated slides, incubated for 15 min with 0.2% NaBH$_4$, washed with TS and blocked with TS containing 5% dry milk powder (TS-M) for 30 min. The slides were incubated with primary antisera (all diluted 1:250 in TS-M) for at least 2 h. After two rinses with TS, slides were incubated for at least 2 h with secondary antibodies diluted 1:300. After two rinses with TS, slides were mounted in antifading medium, which optionally included DAPI (50 ng/ml).

## Immunofluorescence of leaf sections
The 14-μm microtome sections were immobilized on polylysine-coated slides, incubated for 1 h with 0.2% NaBH$_4$, washed with TS and incubated for 90 min in an enzyme solution (containing 2% cellulose 'Onozuka' R10, 1% macerozyme R10) and 2% driselase (http://www.sigmaaldrich.com, prepared in 10 mM MES pH 5.6). The 50-μm vibratome sections were incubated for 1 h with 0.2% NaBH$_4$ in a 24-well plate. All sections were then blocked with 3% BSA or 5% BSA in TS supplemented with 0.01% Tween20 for 30 min and incubated with primary antisera in 1% BSA/0.01% Tween20 in TS-M for at least 12 h at the following dilutions: 1:200 for rabbit anti-P2 and rabbit anti-P4 and 1:100 for mouse anti-α-tubulin. After two rinses with TS or PBS, slides were incubated for at least 12 h with Alexa Fluor conjugates at a 1:200 dilution. After two rinses with TS or PBS, slides were mounted as described above.

## Microscopy
Slides were observed with Zeiss LSM510 or LSM700 (http://www.zeiss.com) or Leica SP2 (http://www.leica.com) confocal microscopes operated in sequential mode to avoid crosstalk. Raw images

were processed using LSM, ZEN or LAS software and final figures were prepared using GIMP 2.6.11 (http://www.gimp.org) and OpenOffice 3.4 (http://www.openoffice.org). Quantification of TB pheno-type was performed by counting 200–300 cells in 10 different, randomly chosen microscopy fields per treatment.

## Transmission tests

Groups of about 500 aphids were placed inside copper rings covered with stretched Parafilm M membranes (http://www.parafilm.com) for a 1 h pre-acquisition period in a humid chamber. Then, either protoplasts or suspensions containing purified virus particles, P2 and P3 in SES buffer (*Blanc et al., 1993a*) were placed on the Parafilm M and covered with a cover slip. Aphid were then allowed a 15 min acquisition feed through the Parafilm membranes on the suspensions. For plant-to-plant trans-mission experiments, aphids were transferred to an infected detached leaf for 1–5 min acquisition feeding. Afterwards, either 1 aphid (fed on a leaf) or 10 aphids (fed on protoplasts or virus particles) were transferred onto each turnip test plantlet for a 4 ± 1 h inoculation period; 24 plants were inocu-lated per plant tray and 12 trays were used in a typical assay. Aphids were killed with 0.2% Pirimor G (http://www.certiseurope.fr) as described in *Martinière et al. (2011a)*. Finally, the fraction of symptom-atic plants was scored by visual inspection 3 weeks later.

## Analyzing movement of GFP-TUA6 in live cells by fluorescence recovery after photobleaching (FRAP)

FRAP experiments were performed according to *Martinière et al. (2011b)* on infected or healthy Arabidopsis *GFP-TU6* leaves using a Zeiss LSM700 confocal microscope with a 63× NA 1.4 oil-immersion objective. Leaf samples were mounted in 1% low melting point agar to prevent focus shift. Twenty scans of the entire field of view were made at pre-bleach intensity, and then a circular 20 µm$^2$ region of interest (ROI), which included a TB, was photobleached. Three iterations of the 488-nm laser at 100% intensity were used for the bleaching. For recovery of the fluorescence in TBs, images were recorded for 110 s, with a 512 × 512 px picture size, a scan speed of 167 ms/frame and a delay between frames of 0.2 s. For controls, to account for fast tubulin diffusion in the cytosol, recovery was recorded in identical ROIs for only 5.9 s, with a 100 × 50 px picture size, a scan speed of 16 ms/frame and a delay between frames of 23 ms. We verified that the energy of the 488-nm laser used for the post-bleach added no bleaching effect by recording a control region outside the bleaching ROI. The experiment was repeated 21 times for wild type TB bleaching, 23 times for TB-TC bleaching and 18 times in the case of cytoplasmic soluble tubulin. Average intensities in all ROIs including the back-ground signal were measured using ImageJ 1.44p software (http://imagej.nih.gov/ij), before exporting data into Microsoft Excel 2007 (http://www.microsoft.com).

Fluorescence recovery data was normalized as follows:

$$I_n = ((I_t - I_{min})/(I_{max} - I_{min})) \times 100,$$

where $I_n$ is normalized intensity, $I_t$ is intensity at any time t, $I_{min}$ is the minimum intensity post bleach and $I_{max}$ is the mean intensity pre-bleach.

Non-linear regression was used to model FRAP data. In this case, a one-phase exponential curve was used:

$$Y_{(t)} = A \, Exp^{(-k)(t)} + B,$$

where A, B and k are parameters of the curve and t is time.

From this curve, the half time of recovery was calculated as $t_{(1/2)} = 0.69/k$. Finally, $t_{(1/2)}$ was used to calculate the diffusion rate as $D = (0.88 \, R^2)/(4 \, t_{(1/2)})$, where D is the diffusion rate and R is the radius of the bleaching area.

## Robostylets (microelectrode experiments)

A pulled glass microelectrode was fitted on a micromanipulator (http://www.prioruk.com) and placed either on the stage of a Leica confocal LSI macroscope equipped with a 0.56–16× zoom and a 5× objective or a Zeiss LSM700 confocal microscope with a 10× objective. Either a whole potted plant (visualized with the macroscope) or a detached leaf taped to a slide and with a water-soaked paper wrapped around its stalk (visualized with the microscope) was placed under the objective. The microelectrode was carefully brought up to the leaf, and the epidermis was touched or pierced. The

approach, the mechanical stress and the reaction of the plant leaf epidermis cell were all recorded by time lapse fluorescence microscopy using acquisition settings as described above for the LSM700 microscope, or excitation with a 488-nm diode laser and an emission bandwidth from 505–550 nm for the LSI macroscope. The pinholes were opened to record sections approximately with 20-µm thickness, and the microscope settings were selected for minimal acquisition times at the expense of image quality.

## Statistical analysis

For FRAP, D values were compared with a two-tailed t-test. TB activation states close to and distant from salivary sheaths, as well as transmission rates, were analyzed using GLM and hierarchical GLM models with a binomial distribution. For P2-TC transmission experiments, the Mann–Whitney test was used. To test for differences in TB states between wild type-infected and P2-TC-infected tissue, a nominal logistic model was used since three parameters were analyzed. Statistical analyses were carried out using JMP 10 (http://www.jmp.com), R 2.9.2 (http://www.r-project.org) and Vassarstats (http://vassarstats.net/) software. The p values <0.05 were regarded as statistically significant.

## Acknowledgements

We thank Gérard Labonne for aphid rearing and Sophie Le Blaye for plant care, Yves Prin for help with histology, and Takii Seed Company for providing turnip seeds. We are also deeply grateful to Takashi Hashimoto for sending Arabidopsis *GFP-TUA6* seeds. Macroscope acquisitions were performed at the MRI imaging platform (Montpellier, France). Thanks to Anouk Zancarini for designing the aphids in *Figures 1 and 11*. The language of the manuscript was proofread and corrected by Brandon Loveall (http://www.improvence.net). AB acknowledges a PhD fellowship financed by the SPE department of INRA and the Région Languedoc-Roussillon; AMa thanks the French government for a PhD fellowship. MD and AMa thank Carl Zeiss, Soichiro Honda and Robert Pirsig for ZEN and the art of motorcycle maintenance.

## Additional information

### Funding

| Funder | Grant reference number | Author |
|---|---|---|
| INRA, Département SPE, France | | Martin Drucker |
| Agence National de Recherche (ANR), France | BLAN07-2-192768 | Stéphane Blanc |

The funders had no role in study design, data collection and interpretation, or the decision to submit the work for publication.

### Author contribution

AMa, Conception and design, Acquisition of data, Analysis and interpretation of data; AB, Conception and design, Acquisition of data, Analysis and interpretation of data, Drafting or revising the article; J-LM, Acquisition of data, Analysis and interpretation of data; NL, Acquisition of data, Analysis and interpretation of data; DG, Acquisition of data, Analysis and interpretation of data; JD, Analysis and interpretation of data; EG, Acquisition of data; AMo, Acquisition of data; AF, Acquisition of data; SB, Conception and design, Acquisition of data, Analysis and interpretation of data, Drafting or revising the article; MD, Conception and design, Acquisition of data, Analysis and interpretation of data, Drafting or revising the article

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
