## [Decision Letter]

Thank you for choosing to send your work entitled “Perceiving the outside world: a virus uses the host sensory system for instantaneous transmission by an insect vector” for consideration at *eLife*. Your article has been evaluated by a Senior Editor, a Reviewing Editor, and 2 reviewers. The following individuals responsible for the peer review of your submission want to reveal their identity: Thorsten Nürnberger (BRE); Georg Jander (peer reviewer); Saskia Hogenhout (peer reviewer).

The reviewers discussed their comments before we reached this decision. Our goal is to provide the essential revision requirements as a single set of instructions, so that you have a clear view of the revisions that are necessary for us to publish your work.

The discovery that aphid feeding and other stresses can trigger a tubulin-dependent reorganization of transmission bodies in plant cells infected with CaMV, that this reorganization depends on viral protein P2, and that the “mixed networks” mediate viral transmission are original and significant contributions and highly appropriate for *eLife*. However, *eLife* is a broad audience journal and it is critical that the manuscript is written in a manner that will be accessible to its readership. Accordingly:

1. The manuscript should be re-written to provide background information and to improve the writing. Special attention should be given to the Introduction and Results section ‘TB transformation enables efficient transmission’. Consider obtaining detailed feedback from a colleague who is not in the plant virus field.

2. The confocal work should include line scans to show the position of tubulin and EM images are needed for tubulin minus and plus transmission bodies.

3. Label the figures in a manner that is more intuitive with each micrograph separately labeled and provide clear explanations for some of the figures. Specifically, the graphs in Figure 3B need to be explained, Figures 2, 3, 4, and 5K need improved labeling, and Figure 6C was difficult to interpret.

4. Figure 1 should be expanded to provide an overall model for the paper with a timeline of key events. Consider moving it to the end.

5. We would ask you to re-phrase the title and abstract of your study. In our view, this is mandatory as it indicates that viruses indeed make use of the host machinery for vector-mediated transmission. While this hypothesis seems plausible, it was not investigated in your study to an extent that would justify this rather strong statement.

---

## [Author Response]

*1. The manuscript should be re-written to provide background information and to improve the writing. Special attention should be given to the Introduction and Results section ‘TB transformation enables efficient transmission’. Consider obtaining detailed feedback from a colleague who is not in the plant virus field*.

We took great care to present our results in a way appropriate for a broad audience. Many parts of the manuscript were rewritten (especially the Introduction, the Results section, and the figure legends). The text was edited for language and readability by a native English-speaking biologist not from the plant virus field. We hope that the text is now easier to understand.

*2. The confocal work should include line scans to show the position of tubulin and EM images are needed for tubulin minus and plus transmission bodies*.

Line scans were added to Figure 2A–C and the new Figures 5A,B and 6A–C. This was a cool idea because the profiles show clearly the differential distribution of P2 within a TB (TB cortex with strong P2 label and center region with weaker P2 label and the opposite for the a-tubulin label). The text was adapted accordingly.

*3. Label the figures in a manner that is more intuitive with each micrograph separately labeled and provide clear explanations for some of the figures. Specifically, the graphs in Figure 3B need to be explained, Figures 2, 3, 4, and 5K need improved labeling, and Figure 6C was difficult to interpret*.

The original 6 figures were split up into 11 newly designed figures. We took care to improve the labeling: where possible, each panel has now its proper character label.

*4. Figure 1 should be expanded to provide an overall model for the paper with a timeline of key events. Consider moving it to the end*.

To leave some “suspense” we explain the biological model in Figure 1 as before, and present the different steps of TB activation in the new Figure 11 that summarizes the results.

*5. We would ask you to re-phrase the title and abstract of your study. In our view, this is mandatory as it indicates that viruses indeed make use of the host machinery for vector-mediated transmission. While this hypothesis seems plausible, it was not investigated in your study to an extent that would justify this rather strong statement*.

To take account of your remarks on the title, we changed it to “Looking outside: A virus responds instantly to the presence of the vector on the host and forms transmission morphs”. We believe that this title better reflects our work and is less speculative. If you have any suggestions for alternative titles, we will be happy to consider them. The Abstract, the Digest and the impact statement were also changed to respect the fact that, although the results suggest it, we do not directly prove viral use of the host sensory system.